# Shift in cold-point tropopause trends derived from radiosonde, satellite, and reanalysis data

Mona Zolghadrshojaee[1], Susann Tegtmeier[1], Sean M. Davis[2], Robin Pilch Kedzierski[3,4], Leopold Haimberger[4]

[1]Institute of Space and Atmospheric Studies, University of Saskatchewan, Saskatoon, Canada
[2]National Oceanic and Atmospheric Administration Chemical Sciences Laboratory, Boulder, CO 80305, USA
[3]Meteorological Institute, University of Hamburg, Hamburg, Germany
[4]Department of Meteorology and Geophysics, University of Vienna, Vienna, Austria

*Correspondence to*: Susann Tegtmeier (susann.tegtmeier@usask.ca)

**Abstract.**

The tropical tropopause layer is the transition region between the well-mixed convective troposphere and the radiatively controlled stratosphere and plays a crucial role for air mass transport between these layers. In this paper, we present updated trends of tropopause and lower stratospheric temperature from radiosondes, GNSS-RO data and the reanalyses ERA5, JRA-3Q, and MERRA-2. Given its importance in determining the concentration of water vapor entering the stratosphere, we focus on temperature trends at the cold-point tropopause, which we determined from radiosonde observations after correcting for time-varying biases.

Radiosonde and GNSS-RO data show a significant shift from strong cold-point cooling for 1980-2001 to warming for 2002-2023. Reanalysis data sets reproduce the robust change in the tropical tropopause temperature trends and furthermore show opposite trends in tropical upwelling for 1980-2001 compared to 2002-2023, consistent with the cold-point and lower stratosphere temperature changes. The shift in cold-point temperature trends around 2000 suggests a regime shift in the dominant mechanism controlling CPT temperatures, from ozone depleting substance-induced dynamical changes before 2000 to greenhouse gas-induced radiative warming with some dynamical contributions after 2000. While the role of dynamical changes after 2000 is not completely clear, this regime shift suggests that in the absence of strong dynamically induced cooling trends, radiative warming could dominate the cold-point temperature trends and thus stratospheric water vapor entry values.

## 1 Introduction

The tropopause, acting as a boundary between the troposphere and stratosphere, plays a crucial role for air mass transport between these layers. Its variations can significantly impact atmospheric dynamics, circulation, and the distribution of greenhouse gases (GHGs) in the upper troposphere and lower stratosphere (UTLS, Holton et al., 1995; Stohl et al., 2003). Therefore, quantifying the tropopause's climatological features and trends is vital for understanding and simulating these

broader atmospheric processes. In the tropics, the cold-point tropopause (CPT) is identified as the point of minimum temperature in the vertical temperature profile (Highwood and Hoskins, 1998). The CPT is particularly important for stratospheric composition and regulating water vapor entry into the stratosphere, making it essential to study both its properties and variations.

Numerous studies have examined the variability and trends of the tropopause layer using various data sources, including radiosonde measurements, satellite observations, reanalysis data, and climate models (Randel et al., 2006; Seidel and Randel, 2006; Rosenlof and Reid, 2008; Wang et al., 2012; Randel et al., 2017; Zolghadrshojaee et al., 2024). These studies have contributed to our understanding of tropopause behavior and its implications for atmospheric processes. Focusing on the time period before mid-2000s, studies have documented a long-term cooling of the cold-point temperature. Rosenlof and Reid

(2008) reported a cooling trend at the cold point and at 100 hPa between 1980 and 2003, with temperature decreases ranging from −0.5 to −1 K per decade, based on radiosonde observations from 52 stations. This finding aligns with Zhou et al. (2001), who observed a cooling trend of about −0.57 ± 0.06 K per decade of the CPT from 1973 to 1998 using operational sounding data. Focusing on the tropical region (15°S-15°N), Seidel et al. (2001) analyzed data from 83 radiosonde stations for the period 1978-1997. They found that the tropical tropopause height increased by 20 meters per decade, while its temperature decreased

by −0.5 K per decade. The reliability of these observations is supported by studies comparing different measurement techniques. Anthes et al. (2008) and Ho et al. (2017) demonstrated strong consistency between temperature profiles obtained from the Global Navigation Satellite System – Radio Occultation (GNSS-RO) and those from radiosondes, lending credibility to the observed trends. However, Wang et al. (2012) demonstrated that adjusted radiosonde temperatures give smaller cooling trends for 1979-2005 with larger uncertainties than the unadjusted data sets. In addition to the overall cooling trend before the

early 2000s, tropopause temperatures appear to have dropped abruptly by ~1.5° to unusually and persistently low values in the years after 2000, consistent with a similar drop in stratospheric water vapour (Randel et al., 2006; Rosenlof and Reid, 2008). Recent studies have revealed a shift in tropical tropopause temperature trends. In contrast to earlier cooling trends, Wang et al. (2015) observed a warming trend in the tropical tropopause layer (TTL) from 2001 to 2011. Their analysis of GNSS-RO data showed a temperature increase of 0.9 K per decade. Similarly, Randel and Park (2019) identified warming of

the CPT from 2005 to 2017, consistent with increased water vapor in the tropopause region as detected by NASA's Aura satellite Microwave Limb Sounder (MLS)onboard the NASA's Aura satellite. Extending the analysis of GNSS-RO data to 2002-2022, Zolghadrshojaee et al. (2024) confirmed the cold point warming, with the strongest warming occurring during the boreal winter and spring reducing the amplitude of the cold-point temperature and water vapour seasonal cycles. The above findings suggest a significant change in tropopause behavior: cooling trends from the 1970s and 1980s until the early 21st

century, followed by a transition to warming in recent years. The representation of changes of tropical tropopause temperature in meteorological reanalyses is of interest for studies of stratospheric transport, composition, and long-term changes. Reanalysis data sets are widely used in such scientific studies, often as "stand-ins" for observations, when the available measurements lack the spatial or temporal coverage needed. In particular, reanalysis products in the TTL region are important

for transport and composition studies with offline chemical transport models or Lagrangian particle dispersion models (e.g., Chipperfield, 1999; Krüger et al., 2009; Schoeberl et al., 2012; Tao et al., 2019). The representation of the cold point in the reanalysis data set used to drive these models determines how realistically such models simulate dehydration and stratospheric entrainment processes.

Reanalysis applications can be complicated by spurious changes and discontinuities in the reanalysis fields due to changes in the quality and quantity of the observations used as input data or the joining of different execution streams (e.g., Fujiwara et al., 2017). A comparison of reanalysis products available at the end of the 1990s with other climatological data sets showed notable differences in temperatures near the tropical tropopause (Randel et al., 2004). Reanalysis cold-point temperature and height data used by Gettelman et al. (2010) for the evaluation of model results showed a considerable spread of the identified cooling trends. Advances of the reanalysis and observational systems over the last decades have led to a clear improvement in the TTL reanalysis products over time with smaller biases and better agreement of the variability from 2006 onward, when densely sampled radio occultation data started being assimilated by the reanalyses (Tegtmeier et al., 2020). The authors reported that most reanalyses suggest small but significant cold-point cooling trends of −0.3 to −0.6 K per decade for 1979-2005 that are statistically consistent with trends based on the adjusted radiosonde data sets.

We aimed to update existing studies of the variability and trends of tropopause temperatures from various observational and reanalysis datasets by covering an extended period from 1980 to 2023. Our analysis focuses on providing adjusted cold-point temperature trends based on radiosonde data for the 44-year long time period. Detailed analysis of the seasonal cycle and spatial patterns of the temperature trends are given for the more recent 2002-2023 time period based on GNSS-RO data. Reanalysis temperature trends will be evaluated for all the above diagnostics. Compared to our previous study on temperature and water vapor trends from 2002-2022 (Zolghadrshojaee et al., 2024), this work will focus on shifts of temperature trends over the full 44-year long time period and the reliability of trend estimates from reanalyses. Sections 2 and 3 describe the datasets utilized and the applied methodology. In Sect. 4, we presented recent temperature trends for 2002-2023, while Sect. 5 focuses on trends over 1980-2001.

## 2 Data

### 2.1 Radiosonde data

We use radiosonde data from different sources to evaluate TTL temperature changes over the past 44 years. First, we use unadjusted, quality-controlled radiosonde data available on all levels reported by the radiosondes. This data is known to contain inhomogeneities or time-varying biases caused by changes in instruments and measurement practices (e.g., Seidel and Randel, 2006), complicating trend estimation. Second, we use adjusted radiosonde data where inhomogeneities were removed based on information from reanalysis data or neighbouring stations. While the adjusted data sets are believed to provide more realistic trend estimates, the calculation of the cold-point suffers from the data being available only at fixed pressure levels (Haimberger, 2007; Haimberger et al., 2008).

**Unadjusted radiosonde data**

We used a monthly mean, gridded temperature dataset at the cold point as well as at the 100 and 70 hPa pressure levels based
on the unadjusted, quality-controlled Integrated Global Radiosonde Archive version 2.2 (IGRA 2.2; Durre et al., 2006) for the
period from 1980 to 2023. IGRA data stems from radiosonde and pilot balloon observations from over 2,800 globally
distributed stations with the first data from soundings in 1905.

For calculating the cold point based on unadjusted IGRA data, we followed the criteria outlined by Wang et al. (2012). For
each sounding at one of the 110 tropical stations, we selected the minimum temperature between 300 hPa to 70 hPa, excluding
soundings that lacked data at any level between. In a second step, monthly mean cold-point temperatures at each station were
calculated for t00 (00:00 UTC) and t12 (12:00 UTC) requiring at least 15 daily recordings within a month for each time
respectively.

Trends of cold-point as well as 70 and 100 hPa temperatures were derived from monthly mean data based on the multiple
linear regression analysis (Sect. 3.1), provided that at least 70% of the data was available. Outliers were identified and removed
using the Interquartile Range (IQR) method. Data points falling outside the 25th and 75th percentiles of the temperature
distribution for each station (less than 1% on average) were considered outliers and excluded from the dataset prior to trend
calculation. Trends were computed for each station individually and then averaged over each grid cell of a 30° longitude × 16°
latitude grid within 30°S to 30°N to obtain mean trends for T00 and T12 separately. In the final step, tropical average (30°S-
30°N) temperature trends were obtained by averaging the trends over all grid cells and both times. The IGRA dataset consists
of observations from 144 stations, some of which have incomplete records due to gaps in their time series. Additionally, the
spatial distribution of stations is uneven, with substantial gaps over the oceans.

**Adjusted radiosonde data**

The accuracy of temperature trends derived from radiosonde data can be impacted by inconsistencies or biases resulting from
changes in instrumentation or measurement methodologies (Seidel and Randel, 2006). To address potential biases in trends
estimated from unadjusted radiosonde data, we utilized two adjusted radiosonde temperature datasets, the Radiosonde
Observation Correction Using Reanalysis (RAOBCORE; Haimberger, 2007), and Radiosonde Innovation Composite
Homogenization (RICH; Haimberger et al., 2008). RAOBCORE adjustments are based on reanalysis used to locate and adjust
temporal discontinuities, while RICH determines adjustments for the same change points by using neighboring stations. These
adjusted gridded data sets (10° longitude × 10° latitude) are available only at widely spaced fixed pressure levels (100 and 70
hPa in the upper TTL), and not at the cold point. The adjusted data sets come at monthly resolutions spanning the period from
1980 to 2023 and do not provide information on individual soundings.

Deseasonalized tropical mean temperature anomalies for the CPT, as well as at the 70 hPa and 100 hPa levels based on IGRA, RAOBCORE, and RICH datasets from 1980 to 2023 are shown in Fig. 1. Anomalies are calculated relative to the mean annual

cycle for 1980-2023 for each data set. The 70 hPa level (Fig. 1a) shows slightly larger temperature variations compared to the other two levels, with anomalies ranging approximately from -2K to +3K. All three datasets show very close agreement of their interannual anomalies, which are dominated by the stratospheric quasi-biennial oscillation (QBO) signal. The situation is very similar at the 100 hPa level (Fig. 1c) with excellent agreement between RAOBCORE and RICH and relatively good agreement of IGRA with the other two. At the cold point, only IGRA data is available and displays interannual anomalies and

long-term changes similar to the signals at 70 and 100 hPa. Positive temperature anomalies following the eruptions of El Chichón in 1982 and Mount Pinatubo in 1991 can be detected at the 70 hPa level but are less evident at the cold point and the 100 hPa level (Fujiwara et al., 2015).

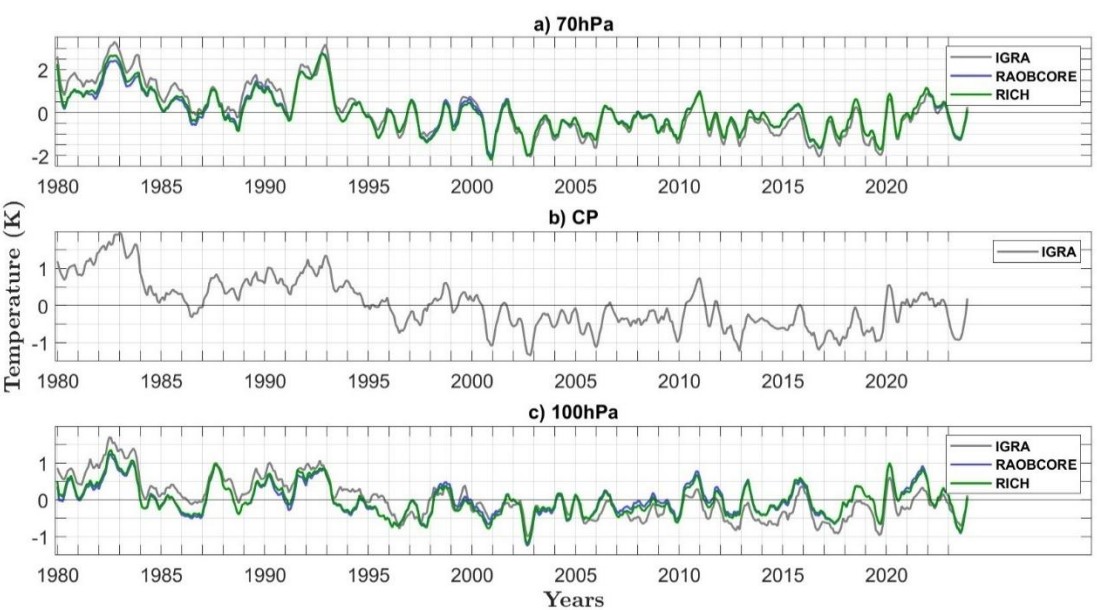


**Figure 1**. Tropical mean (30°S-30°N) deseasonalized temperature anomalies from 1980 to 2023 for (a) 70 hPa, (b) the cold-point tropopause (CPT), and (c) 100 hPa levels, using IGRA (gray), RAOBCORE (blue), and RICH (green) datasets. The data are smoothed over 5 months for visibility.


## 2.2 Satellite data

We utilized GNSS-RO data, which offers global coverage, remains unaffected by clouds and precipitation, and has high vertical resolution and accuracy (Kursinski et al., 1997). In this study, we combined various satellite missions, including Challenging Minisatellite Payload (CHAMP; Wickert et al., 2001), The Constellation Observing System for Meteorology,
Ionosphere, and Climate (COSMIC; Anthes et al., 2008), The Gravity Recovery and Climate Experiment (GRACE; Beyerle et al., 2005), Meteorological Operational satellite (Metop; Von Engeln et al., 2011), successive Metop-B, Metop-C, Argentine Satelite deAplicaciones Cientificas-C (SAC-C; Hajj et al., 2004), Terra Synthetic Aperture Radar - X-band (TerraSAR-X; Krieger et al., 2007; Beyerle et al., 2011), and the KOrea Multi-Purpose SATellite-5 (KOMPSAT-5; Cho et al., 2009; Bowler, 2018). Specifically, we used monthly mean CPT temperature, along with temperature values at fixed pressure levels (30, 50,
60, 70, 100, 150, 200, 300, and 400 hPa). These data are sourced from all GNSS-RO missions and are spatially gridded on a 30° longitude × 10° latitude grid within the latitudinal range of 30°N to 30°S, covering the period 2002-2023. We calculated temperature trends at the cold point and at fixed pressure levels for the gridded data and for tropical mean averaged data following the methodology given in Sect. 3.1.


## 2.3 Reanalysis data

We used temperature data from three reanalysis datasets, the Modern-Era Retrospective Analysis for Research and Applications, v2 (MERRA-2; Gelaro et al., 2017), the European Centre for Medium-Range Weather Forecasts Reanalysis v5
(ERA5; Hersbach et al., 2020), and the Japanese Reanalysis for Three-Quarters of a Century (JRA-3Q; Kosaka et al., 2024) covering the period from 1980 to 2023. Our analysis is based on gridded monthly mean tropical temperature fields extracted at the cold point and at various pressure levels encompassing the TTL and extending into the upper troposphere and lower stratosphere.

Lower stratospheric temperature from ERA5 has a cold bias for the years from 2000 to 2006 as a result of the specified
background error covariances for the data assimilation. The bias was corrected in an updated product (ERA5.1) produced for 2000-2006 which provides improved global-mean temperatures from the stratosphere to the uppermost troposphere (Simmons et al., 2020). Here we used a combined product of ERA5.1 for 2000-2006 and ERA5 for all other years, which we refer to as ERA5 for simplicity.

Conventional observations—including surface measurements, balloon and aircraft data, as well as satellite observations—are
assimilated to constrain the temperature fields in the reanalysis data (e.g., Tegtmeier et al., 2020). All of the above reanalysis systems assimilate microwave and infrared radiance from the satellite sounders of the Television Infrared Observation Satellite Operational Vertical Sounder (TOVS) suite (1979–2006) and the Advanced Television Infrared Observation Satellite Operational Vertical Sounder (ATOVS) suite (1998–present), with the latter improving the vertical resolution of the assimilated data. In addition, the reanalyses assimilate radiance estimates from the hyperspectral Atmospheric Infrared Sounder

(AIRS; 2002–present), Infrared Atmospheric Sounding Interferometer (IASI; 2008–present), and/or Cross-track Infrared Sounder (CrIS; 2012–present). Temperature observations from radiosondes are assimilated by all reanalyses, with systematic errors due to solar heating of the temperature sensor typically being corrected either on-site or at the reanalysis center before assimilation (Fujiwara et al., 2017). By assimilating RAOBCORE (JRA-3Q, MERRA-2) and RICH (ERA5) data discontinuities in temperature time series from unadjusted radiosondes are avoided.

The reanalysis resolve the TTL with different vertical resolutions, with the number of model levels between 200 and 70 hPa varying from 14 (JRA-3Q) and 7 (MERRA-2) to 21 (ERA5) corresponding to vertical resolutions between ~1 and ~0.2 km. The reanalysis cold-point tropopause temperature was computed globally from model-level data at each grid point by taking the minimum temperature between 500 and 10 hPa at the finest temporal resolution available (hourly for ERA5, 3-hourly for MERRA-2, and 6-hourly for JRA-3Q). The data used for the CPT calculation are at the native output horizontal resolution
(0.667° × 0.5° for MERRA-2 and 0.375° × 0.375° for JRA-3Q) except for ERA-5, which was regridded to 1° x 1° due to bandwidth and file size constraints. The reanalysis model-level pressure fields were interpolated to fixed pressure levels (30, 50, 60, 70, 100, 150, 200, 300, and 400 hPa) at each grid point. Zonal and tropical averages of the cold point and pressure level temperature fields were calculated by averaging over all grid points, representing the final step of data processing.

Reanalysis estimates of the residual velocity were taken from the Atmospheric Processes And their Role in Climate (APARC)
Reanalysis Intercomparison Project (A-RIP) zonal mean data set Martineau et al. (2018). In particular, we used the vertical residual velocity, $\overline{\omega}^*$, from the Transformed-Eulerian-mean diagnostics based on the primitive momentum equation. Vertical velocities in reanalysis products are known to be highly dependent on the specific implementation of data assimilation step and less constrained by observations than the meridional velocities. A comparison of the reanalysis vertical residual velocities to the corresponding estimates calculated from the continuity equation revealed reasonable agreement in the troposphere and
lower stratosphere up to 10 hPa (Fujiwara et al., 2024).

## 3. Methods

### 3.1 Multiple linear regression and trend errors bars

We used multiple linear regression to distinguish between TTL temperature long-term trends and short-term variations that
can be accounted for by various atmospheric phenomena. This analysis utilized deseasonalized temperature anomalies at both the CPT and standard pressure levels, using gridded data and tropical averages from the given datasets listed above. By applying the regression described in Eq. (1), we separated the gradual temperature changes over time from those influenced by factors such as QBO, ENSO, and stratospheric aerosol optical depth (SAOD).

$$T(t) = A + B \cdot t + C_1 \cdot QBO_1(t) + C_2 \cdot QBO_2(t) + D \cdot ENSO(t) + E \cdot SAOD(t) + \epsilon(t), \qquad (1)$$

where $T(t)$ are the temperature anomalies and $\epsilon(t)$ are the residuals. The output of the regression provides the linear trend in the form of the regression coefficient $B$. The $QBO_1(t)$ and $QBO_2(t)$ terms are orthogonal time series representing QBO

variations constructed as the first two empirical orthogonal functions of the Freie Universität Berlin (FUB) radiosonde stratospheric winds (Naujokat, 1986; Wallace et al., 1993). $ENSO(t)$ is the multivariate ENSO index (https://www.esrl.noaa.gov/psd/enso/mei/), and $SAOD(t)$ is the detrended stratospheric aerosol optical depth from the Global Space-based Stratospheric Aerosol Climatology (GloSSAC; Kovilakam et al., 2020).

We calculated the uncertainty in each long-term trend as the standard error of the slope with the effective sample size adjusted to account for the corresponding lag-1 autocorrelation coefficient as described in Santer et al. (2000). Significance was tested based on a two-tailed test with a 95 % confidence interval.

## 3.2 'Nearby Level' approach

Since the adjusted radiosonde datasets lack cold-point temperature, we employed the methodology described by Wang et al. (2012) to generate CPT temperature trends for these datasets. Specifically, we applied the 'Nearby Level' approach, which compares temperature trends at the CPT and nearby fixed pressure levels in unadjusted data with trends at those fixed levels in the two adjusted datasets. This approach makes use of the fact that inhomogeneities at the cold point usually occur at the same time as those at the surrounding levels as discussed by Wang et al. (2012). Therefore, the trend of the time series of differences can be assumed to be mostly independent of the inhomogeneities. In addition, the 'Nearby Level' approach assumes that the CPT trend can be inferred from the adjusted trend at a specific pressure level.

In the first step, the tropical mean temperature trends were calculated for unadjusted data at the cold point, 100 and 70 hPa, and for adjusted data at 100 and 70 hPa. In the second step, we applied the Nearby Level approach by combining the fixed level trend (using adjusted data) with the trend in the differences between cold point and fixed level (using unadjusted data), as described in Eq. (2).

$$Trend_{CP,adj} = Trend_{fixed,adj} + Trend(T_{CP,unadj} - T_{fixed,unadj}), \qquad (2)$$

where $Trend_{fixed,adj}$ is the trend of adjusted datasets (RAOBCORE or RICH) at the fixed pressure level (70 or 100hPa), while $T_{CP,unadj}$ and $T_{fixed,unadj}$ are the unadjusted temperatures (IGRA) at the cold point and 70 or 100 hPa, respectively.

Figure 2 shows the tropical average temperature trends from 1980 to 2023 before and after applying the Nearby Level approach. The adjusted radiosonde datasets suggest statistically significant cooling at the lower stratospheric level of 70 hPa of around -0.39 ± 0.15 K per decade (Fig. 2a). At the 100 hPa level, only very weak and non-significant cooling trends are derived from the adjusted radiosondes. The unadjusted IGRA data suggests stronger cooling at both levels, most likely due to spurious cooling introduced by effects such as the solar heating of the temperature sensors. These stronger cooling signals are also apparent in the anomaly time series (Fig. 1), where IGRA shows more positive anomalies at the beginning of the time period and more negative anomalies at the end of the time period when compared to the other two data sets.

Given that the cold-point trend based on unadjusted IGRA data (Fig. 2a) is not reliable, the Nearby Level approach was applied to derive adjusted cold-point temperature trends (Fig. 2b) ranging from -0.18 ± 0.10 K per decade (based on 100 hPa RAOBCORE) to -0.21 ± 0.16 K per decade (based on 70 hPa RICH data). The good agreement of the four trend estimates

based on adjusted data highlights the consistency of our method when applied to different pressure levels or data sets. These adjusted trends are only half of the cooling trend suggested by unadjusted data, reaffirming that inhomogeneities in the latter can impact the trend estimates.

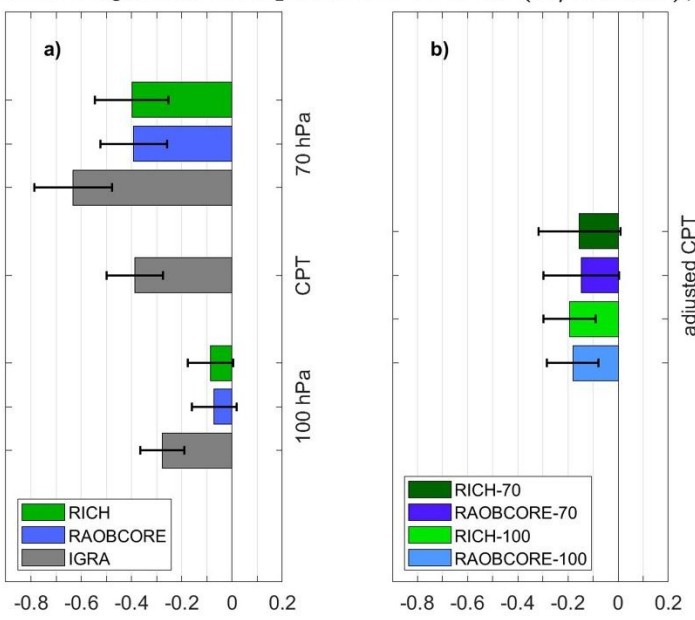

**Figure 2**. Tropical average (30°S-30°N) temperature trends during 1980-2023 for (a) the cold-point tropopause (CPT) and nearby pressure levels for unadjusted (IGRA) and adjusted (RAOBCORE, RICH) data and (b) for the CPT trends based on adjusted data and the Nearby Level approach. Error bars show the ± 2 sigma confidence intervals of the trend taking into account the adjusted effective sample size.

## 4. Temperature trends for 2002 to 2023

Cold-point temperatures experienced a drop-like decrease around the year 2000 and remained anomalously cold over the
following years (e.g., Randel et al., 2006). This feature is also known from stratospheric water vapor changes, which are highly correlated with cold-point tropopause temperatures. GNSS-RO temperature data became available from 2002 onward allowing for more detailed studies of TTL temperature features and trends. In order to avoid the impact of the temperature drop in 2000 on trend estimates and to make use of the high-resolution, dense GNSS-RO data, we first focused our analysis on the 2002-2023 time period and compared the various observational and reanalysis data sets. Figure 3 shows the tropical mean (30°S-
30°N) CPT temperature anomalies from observations (GNSS-RO) and reanalysis (MERRA-2, JRA-3Q, and ERA5). The time series shows large amounts of inter-annual variability, which is well captured across the reanalysis datasets when compared to the satellite data. Agreement among the reanalyses and between the reanalyses and the observations generally improved in 2006 with the beginning of the assimilation of COSMIC GNSS-RO data. Despite the large inter-annual fluctuations, a general

warming trend is evident. Note that the variances of the temperature time series explained by QBO, ENSO or SAOD multiple linear regression terms are relatively small at the cold point with R-square values of less than 0.07 (see Fig. A1). The consistency across all four datasets strengthens the conclusion that tropical CPT temperatures have been increasing over the past two decades, highlighting the importance of ongoing atmospheric monitoring and analysis.

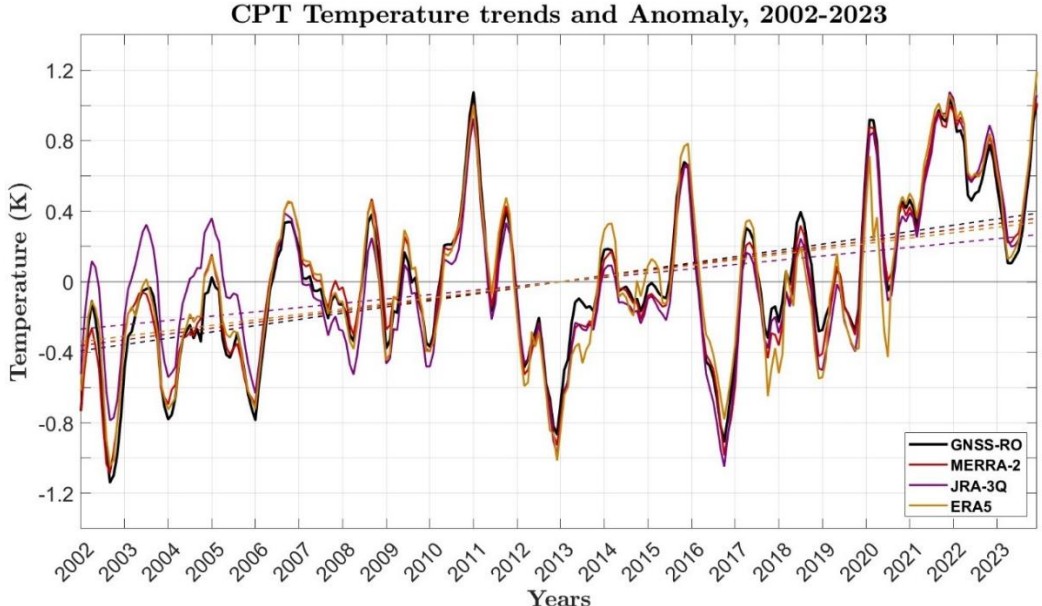

**Figure 3**. Tropical mean (30°S-30°N) deseasonalized cold-point tropopause (CPT) temperature anomalies and the linear trend based on the GNSS-RO, MERRA-2, JRA-3Q, and ERA5 data. The data are smoothed over 5 months for visibility.

The cold-point temperature trends for 2002-2023 for the three reanalysis and GNSS-RO data together with the radiosonde trends are shown in Fig. 4. The reanalysis trends agree very well with the GNSS-RO trend (Fig. 4c), with all four data sets showing significant warming which is strongest for MERRA-2 (0.33 ± 0.17 K per decade) and weakest for JRA-3Q (0.25 ± 0.19 K per decade). These warming trends seem to be driven by the appearance of particularly low-temperature anomalies at the beginning of the time period (2002 and 2006) and particularly high-temperature anomalies at the end of the time period (2020-2023) as illustrated in Fig. 3. Note that trends based on a conventional linear regression with only a constant and a linear term are very similar (with differences of less than 10%) to the trends based on the multiple linear regression (see Fig. A2).

The small differences of the reanalysis trends among each other and with the GNSS-RO trends can have various reasons including differences in the forecast model, assimilation scheme, execution streams as well as quality and quantity of the observations used as input data (e.g., Long et al., 2017; Tegtmeier et al., 2020). All three reanalyses assimilate upper-air temperature observations from radiosondes, but the removal of systematic errors in radiosonde profiles before assimilation differs between the reanalysis centers (Fujiwara et al., 2017). While radiosonde temperatures in JRA-3Q are bias-corrected

with RISE v1.7.2 (Kosaka et al., 2024), ERA5 applies a RICH based homogenization and an additional solar-elevation-dependent correction only up to 2015, then switched to bias adjustments used operationally at ECMWF (Hersbach et al., 2020).

The adjusted radiosonde trends (Fig. 4b) also suggest warming, however with non-significant trend estimates. All four estimates are very similar ranging from 0.14 ± 0.27 K per decade (based on 100 hPa RAOBCORE data) to 0.23 ± 0.39 K per decade (based on 70 hPa RICH data). These trends are about half as strong as the satellite-based warming trend (GNSS-RO) of 0.36 ± 0.16 K per decade consistent with results from Steiner et al., (2020) for the time period 2002-2018. Sampling the reanalysis according to the radiosonde station network does not change the reanalysis trend estimates (not shown here) implying that the limited sampling of the radiosonde network is not the reason for the smaller trend estimates. Potential reasons for the differences between the observational trend estimates are the radiosonde homogenization approaches that rely among other things on differences between radiosonde temperatures and background forecasts (Haimberger et al., 2012).

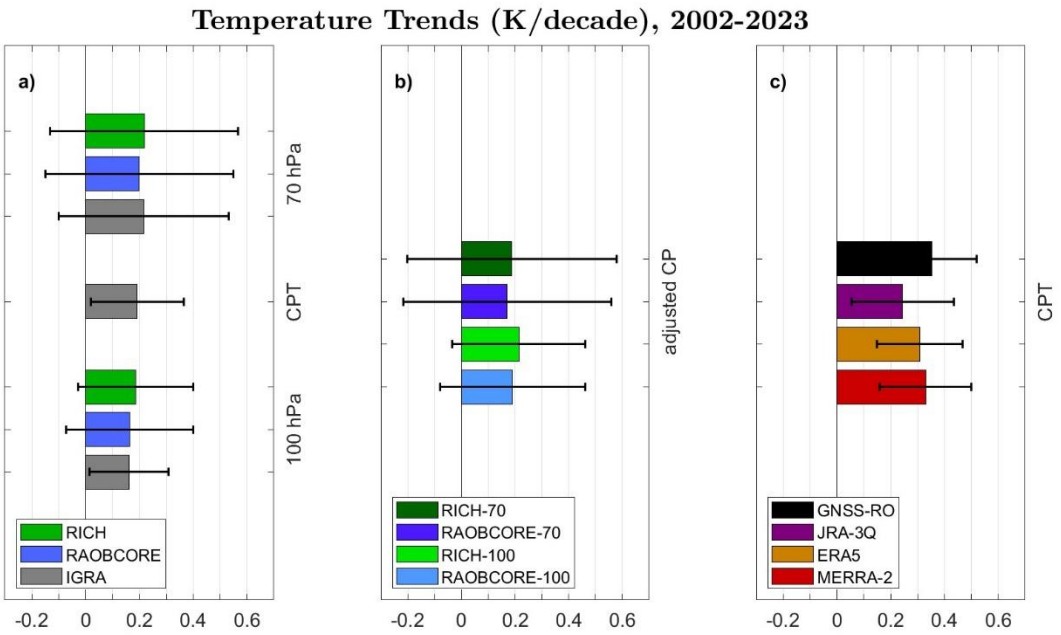

**Figure 4**. Tropical average (30°S-30°N) temperature trends during 2002-2023 for (a) the cold-point tropopause (CPT) and nearby pressure levels for unadjusted (IGRA) and adjusted (RAOBCORE, RICH) data; (b) the CPT trends based on adjusted data and the Nearby Level approach and (c) the CPT based on GNSS-RO and reanalysis data. Error bars show the ± 2 sigma confidence intervals of the trend taking into account the adjusted effective sample size.

We analyzed the spatial patterns of the cold-point temperature trends using regression analysis applied to monthly temperature time series at each grid point for satellite observations and reanalysis (Fig. 5). All datasets show a general warming trend in CPT temperatures across the tropics. Pronounced warming can be seen across all data sets over the tropical North Atlantic

Ocean, tropical South Pacific Ocean, and South Indian Ocean. Temperature trends are insignificant over the northern edge of the tropical regions. Overall, the trends show a latitudinal gradient with stronger warming in the SH tropics. We have tested the impact of the less dense sampling patterns during the earlier GNSS-RO missions such as CHAMP and GRACE on the spatially resolved temperature trends by subsampling the GNSS-RO data. For each month of 2006-2024, we have randomly chosen 6000 profiles globally from all missions to match the sampling density of the earlier GNSS-RO missions. The spatially resolved cold-point trends based on the subsampled data are very similar to the original trends based on unsampled data shown in Figure 5a (for a direct comparison see Fig. A3). While some smaller differences can be seen, the magnitude and distribution of the trends show overall excellent agreement suggesting that the varying sampling density only has a very small impact on the results presented in our manuscript.

The spatial patterns of the seasonal mean trends are presented in Appendix A (Figs. A4-A7), offering a detailed view of how the cold-point temperature trends vary with season. The analysis shows warming across all seasons, with especially pronounced warming over the western hemisphere in boreal winter, the equatorial region during boreal spring, and the southern tropics during boreal autumn. The latitudinal gradient of the temperature trends results from the trend distributions in boreal summer and autumn. Overall, the reanalysis datasets align well with observations in terms of magnitude and spatial patterns of the cold-point warming that occurred over the last 20 years.

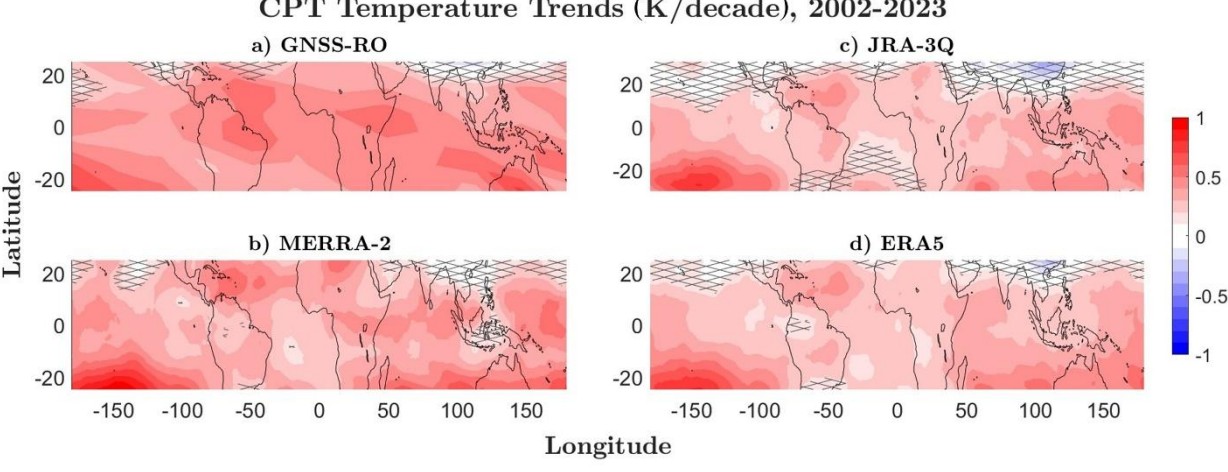

**Figure 5**. Long-term (2002–2023) temperature trends at the cold-point tropopause (CPT) based on (a) GNSS-RO, (b) MERRA-2, (c) JRA-3Q, and (d) ERA5 data. Trends not significant at the 95% confidence level are marked with grey cross-hatching.

The investigation of the seasonal cycle and the vertical structure of TTL temperature trends over 30°N–30°S highlights discernible patterns across the satellite and reanalysis datasets (Fig. 6). Overall, all datasets exhibit approximately the same

behavior. Data from observations indicate substantial warming exclusively between July and December, occurring only at altitudes lower than 150 hPa. In contrast, reanalysis data highlight a broader upper-tropospheric warming trend that extends into the lower stratosphere, reaching up to 80 hPa during boreal spring and summer, 50 hPa in boreal winter, and as high as 30

330   hPa in October. Conversely, stratospheric cooling is strongest during boreal summer in all data sets. In the upper troposphere, the seasonal cycle of the temperature trends is nearly opposite to that of the lower stratosphere, with the largest warming occurring during boreal late summer and fall. An exception to these opposite seasonal cycles is the October warming which is evident across all altitudes.

Temperature trends at the cold point show a clear seasonal cycle, with two pronounced features. First, warming maximises

335   during boreal winter and spring (Zolghadrshojaee et al., 2024), which is especially evident in the MERRA-2 and ERA5 datasets. This strongest cold-point warming during boreal winter could be related to dynamical changes such as a pronounced weakening of the BDC during this time of year. In addition to the boreal winter, the cold point shows a strong warming signal in October, similar to the lower stratosphere, in all data sets. This warming signal might also be related to the weakening of the BDC, as it coincides with a strong negative temperature trend in the Southern Hemisphere polar regions during this month

340   (Ladstädter et al., 2023). Note that the monthly cold-point trends are not significant in most datasets. Overall, the reanalyses reproduce the observed temperature trends very well in terms of their seasonal and vertical structure. The best agreement is found for JRA-3Q, with the other data sets showing slightly more pronounced warming trends in the upper TTL and lower stratosphere.

345

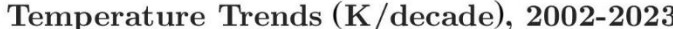

## Temperature Trends (K/decade), 2002-2023

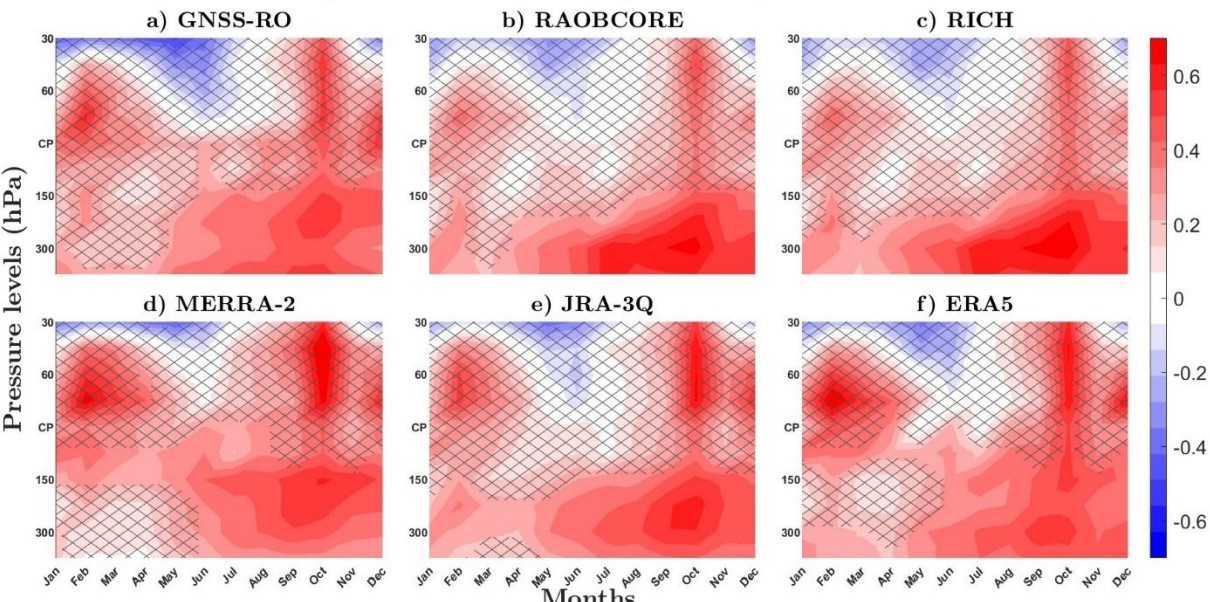

**Figure 6**. The seasonal cycle of long-term temperature trends (2002-2023) at the cold point (CP) and different pressure levels based on (a) GNSS-RO, (b) RAOBCORE, (c) RICH, (d) MERRA-2, (e) JRA-3Q, and (f) ERA5 data (30°N–30°S). Trends not significant at the 95% confidence level are marked with hatches.

350

The spatial patterns of temperature trends at the cold point are known to be correlated with temperature trends in the lower stratosphere and anticorrelated with temperature trends in the upper troposphere (e.g., Lin and Emanuel, 2024; Zolghadrshojaee et al., 2024). Here we tested if the reanalysis data sets show this well-known characteristic by calculating the Pearson correlation coefficients between the gridded trends at the cold point and at all other levels (Fig. 7). The CPT temperature trend patterns exhibit striking similarities with temperature trend patterns at levels slightly below (100 hPa) and above (70 hPa), particularly during the boreal winter and fall, for all data sets. These correlations can extend up to 60 and even 50 hPa, depending on season and data set, with the strongest coupling to higher levels found for ERA5.

Local tropospheric temperature trends are anti-correlated with cold-point (and therefore lower stratospheric) temperature trends via regional dynamical or radiative processes (e.g., Kuang and Bretherton, 2004; Lin and Emanuel, 2024). Significant negative correlations are observed consistently at the 300 hPa and 400 hPa levels with the exception of the boreal spring season (Fig. 7). GNSS-RO, JRA-3Q, and ERA5 datasets exhibit stronger negative correlations compared to MERRA-2 most of the time. There is also some seasonal variability, with stronger negative correlations during boreal winter and summer. Despite some differences among the datasets, the overall patterns remain consistent, highlighting a robust vertical structure of temperature trends in the upper troposphere and lower stratosphere.

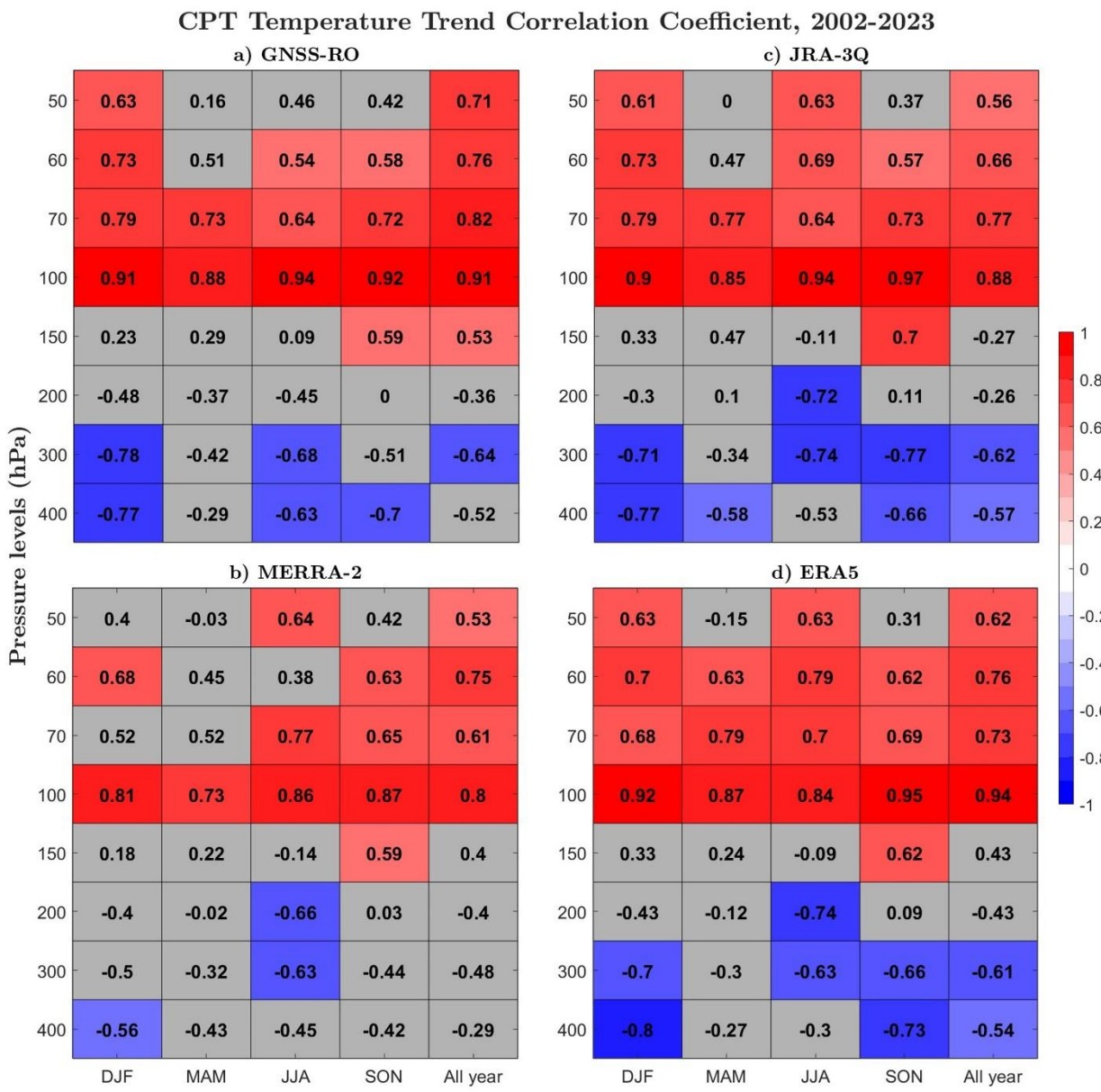

**Figure 7.** Correlation coefficients between gridded temperature trends at the cold-point tropopause (CPT) and the different pressure levels as given on the y-axis for (a) GNSS-RO, (b) MERRA-2, (c) JRA-3Q, and (d) ERA5 for 2002–2023 (30°N–30°S). Correlation coefficients in grey boxes are not significant at the 95% confidence level.

## 5. Temperature trends for 1980 to 2001

The tropical cold-point tropopause experienced significant cooling before the year 2000 (e.g., Zhou et al., 2001; Seidel et al., 2001; Wang et al., 2012) and here we evaluated if the reanalysis data sets reproduce the magnitude and seasonal variations of

these cooling trends. Figure 8 illustrates the tropical average temperature trends for 1980-2001 based on radiosonde data and reanalysis. Unadjusted and adjusted radiosonde data show increasing cooling with altitudes between 100 and 70 hPa, with nearly all trend estimates being significant. The Nearby Level approach results in cooling trends ranging from -0.5 ± 0.31 K per decade to -0.30 ± 0.46 K per decade with the 100 hPa based trend estimates being significant while the 70 hPa based estimates show slightly lower values and larger error bars resulting in non-significant trends (Fig. 8b). Reanalyses also exhibit cold-point cooling, though the trends are notably weaker compared to observations, particularly in the case of MERRA-2.

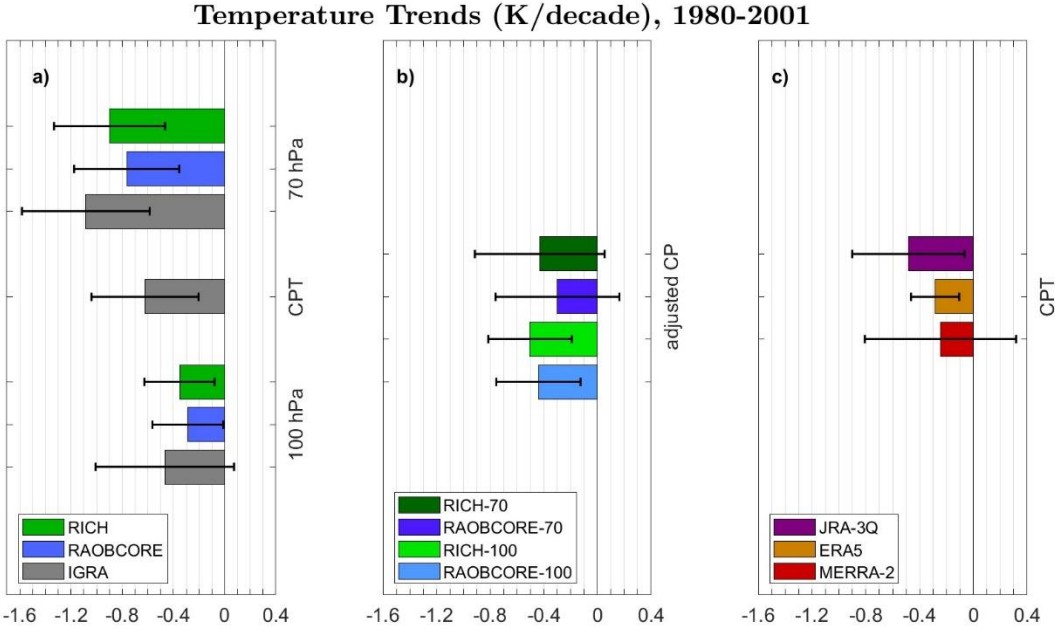

**Figure 8**. Tropical average (30°S-30°N) temperature trends during 1980-2001 for (a) the cold-point tropopause (CPT) and nearby pressure levels for unadjusted (IGRA) and adjusted (RAOBCORE, RICH) data; and (b) the CPT trends based on adjusted data and the Nearby Level approach and (c) the CPT trends based on reanalysis data. Error bars show the ± 2 sigma confidence intervals of the trend taking into account adjusted effective sample size.

Figure 9 illustrates the seasonal cycle and vertical structure of TTL temperature trends over the 30°N–30°S region, based on RAOBCORE, RICH, MERRA-2, JRA-3Q, and ERA5 data for the period 1980–2001. Comparing the adjusted radiosonde trends to the same analysis for 2002-2023 (Fig. 6) reveals smaller warming rates in the upper troposphere (300-400 hPa) for the earlier period reaching only values of up to 0.3 K per decade instead of 0.6 K per decade. This feature of temperature trends being less positive/more negative for the 1980-2001 time period is found at all levels with the differences increasing above the

level of 150 hPa. In the lower stratosphere the differences maximize with cooling trends of around -0.75 K per decade for 1980-2001 compared to a warming trend of around 0.2 K per decade for 2002-2023.

The seasonal cycle of the temperature trends shows that the 1980-2001 lower stratospheric cooling is only interrupted in March when temperature trends are around zero for the radiosonde data. The absence of cooling in March might be related to the weakening of the BDC in this month, as the Arctic polar vortex has intensified during NH spring over 1979-1999 (Langematz and Kunze, 2006). This signature of the BDC trends in the seasonal cycle of the tropical cold-point temperature trends with a reduced cooling trend in March (end of NH winter) is similar to the strong October warming found for 2002-2023 at the end of the SH winter (Fig. 4). All three reanalyses reproduce the seasonal and height variations of the observed temperature trends. Upper tropospheric warming is overestimated by MERRA-2 and also somewhat by JRA-3Q. Similarly, both data sets show slightly weaker cooling at the stratospheric levels. ERA5 temperature trends are most similar to the RAOBCORE and RICH temperature trends.

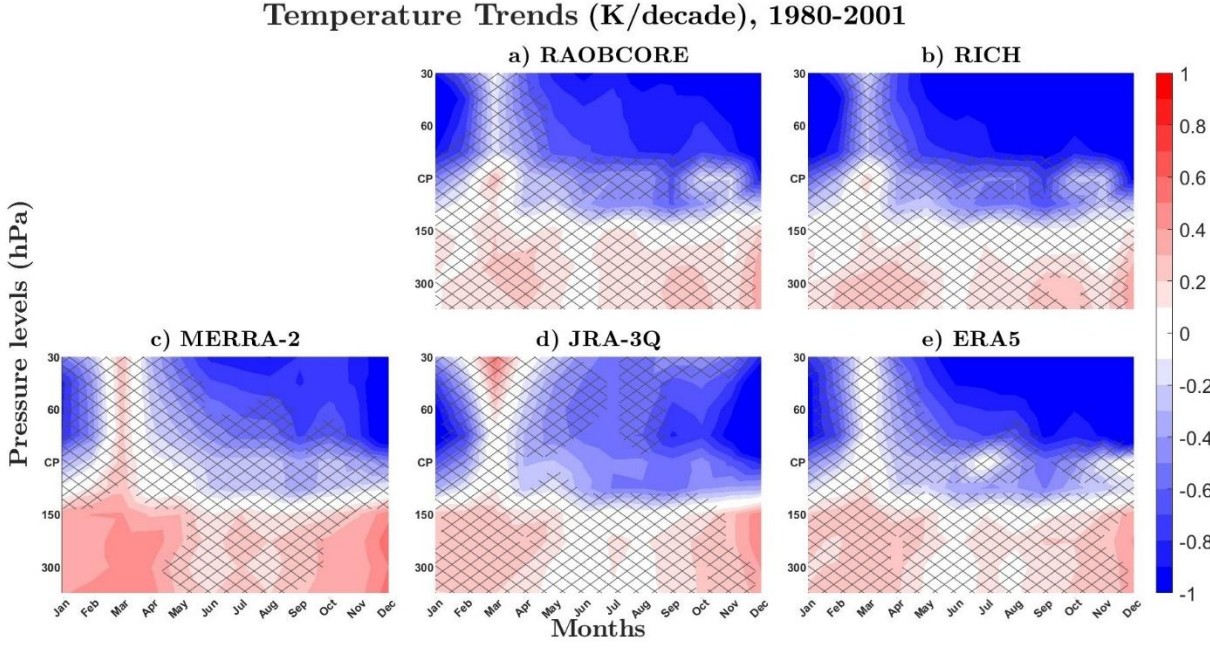

**Figure 9**. The seasonal cycle of long-term temperature trends (1980-2001) at the cold point (CP) and different pressure levels based on (a) RAOBCORE, (b) RICH, (c) MERRA-2, (d) JRA-3Q, and (e) ERA5 data (30°N–30°S). Trends not significant at the 95% confidence level are marked with hatches.

## 6. Discussion

We have analysed updated trends of TTL and lower stratospheric temperature from radiosonde, GNSS-RO and reanalysis data sets. Results are summarized in Fig. 10, which shows the linear trends in tropical average temperature at the CPT across various observational and reanalysis datasets for the periods 1980-2001, 2002-2023, and 1980-2023. The extended period from 1980-2023 displays general cooling trends of up to -0.33 ± 0.12 K per decade (based on JRA-3Q data). While the 1980-2023 cooling is less pronounced than cooling before 2000 (for the 1980-2001 period), the uncertainties are smaller resulting in significant

trends for the observational data sets. Given that the radiosonde observations underestimate the warming trends for the last two decades when compared to GNSS-RO, the cooling trends for the full time period might be closer to zero than implied by the radiosonde time series. Cooling in JRA-3Q is larger than in observations, while MERRA-2 and ERA5 have slightly smaller and not significant cooling rates.

       Warming trends over 2002-2023 are only significant for GNSS-RO satellite observations and the reanalysis data. The latter
reproduces not only the magnitude of the warming trends but also the spatial distributions and seasonal variations. Similar to the observations the reanalysis tropopause trends display anticorrelations with patterns of tropospheric temperature trends, and regions of strongest cold-point warming are found to show slight cooling trends in the upper troposphere.

       Overall, the comparison in Fig. 10 highlights a significant shift in temperature trends from cooling in the earlier period (1980-2001) to warming in the later period (2002-2023) across all pressure levels and datasets, indicating a robust change in the
tropical tropopause temperature dynamics post-2002 (Tao et al., 2023). These changes are consistent with temperature trend changes in the lower stratosphere, where pronounced cooling trends observed before 2000 disappeared and turned into warming trends after 2000.

       Temperature trends in the TTL are driven by increasing GHGs, changing abundances of ozone-depleting substances (ODSs) and their combined effects on composition, dynamics and radiation (Arblaster et al., 2014 and references therein). Well-mixed
GHGs cause a small radiative cooling trend in the lower stratosphere that tends to go to zero with decreasing altitude and turns positive at some level in the upper TTL (Figure 5, Shine et al., 2003). The effects of ozone depletion and reduced absorption of solar radiation, on the other hand, show a relative maximum in cooling trends in the lower stratosphere and upper TTL (As shown in Fig. 1 of Shine et al., 2003).

       Model simulations suggested that before 2000, tropical lower stratosphere cooling was caused by ODSs-driven enhanced
upwelling and related decreases in tropical ozone (e.g., Polvani et al., 2017). GHGs are also known to strengthen the stratospheric circulation (Butchart et al., 2006), adding further to the ODSs-driven increase in upwelling. The cold-point tropopause cooling most likely followed the same mechanism resulting in the clear cooling rates found in radiosonde and reanalysis (Figure 10). While GHG-driven changes in radiation are known to be very small in the lower stratosphere, it is possible that they played a slightly bigger role at the cold-point level by somewhat weakening the ozone driven cooling trend.
This would be consistent with models implying that ozone changes caused similar cooling trends in the upper TTL and lower

stratosphere (Shine et al., 2003), while observations suggest smaller cooling in the upper TTL when compared to the lower stratosphere (Figure 9).

After 2000, lower stratospheric temperatures show no trends or weak warming, which seems to be consistent with the decline of halogenated ODSs following the Montreal Protocol and the slow onset of ozone recovery (World Meteorological
Organization (WMO), 2022). Cooling trends only appear above 50 hPa as can be expected from the radiative effects of GHGs changes (Shine et al., 2003). Temperature trends at the cold-point and in the lower TTL are also positive and can be assumed to be at least partially driven by radiative warming due to increasing GHGs. Seasonal variations of the warming signal suggest that dynamically induced changes might also play a role.

In this study, we examined if the reanalysis trends in tropical upwelling are consistent with trends in cold-point temperature
over the three time periods (see Fig. 10). Reanalysis estimates of vertical velocities are known to suffer from numerical noise as they are computed indirectly from horizontal divergence. In consequence, even relatively small assimilation increments applied to the horizontal winds can have a large influence on the vertical velocity estimates (e.g., Uma et al., 2021). However, since no direct observations of tropical upwelling exist, reanalysis estimates of the vertical velocities have been used in many studies.

Figure 10 shows that upwelling from all three reanalyses have opposite trends for 1980-2001 compared to 2002-2023 consistent with the cold point and lower stratosphere temperature trends. Note that negative values of the vertical velocity given in Pa s$^{-1}$ in Figure 10 correspond to increased upwelling. More upwelling in MERRA-2 and JRA-3Q before 2000 is consistent with cold-point cooling over the same time period. This increase in upwelling is also consistent with a model attribution study suggesting that lower stratospheric cooling before 2000 was driven by an ODS-induced increase in upwelling
(Polvani et al., 2017). Upwelling trends after 2000 are close to zero in MERRA-2 and JRA-3Q suggesting that the post-2000 cold-point warming is not driven by upwelling changes but by radiative effects of increasing GHGs or other dynamical processes. For ERA5, shifts in upwelling trends around 2000 are also consistent with changes in cold-point temperature trends. However, they suggest different mechanisms not consistent with model results. Reduced ERA5 upwelling after 2000 implies stratospheric circulation changes as potential driver of the post-2000 cold-point warming, while the zero upwelling trends
before 2000 suggest that pre-2000 cold-point cooling was driven by direct radiative effects.

The circulation trends over the full time period (1980-2023) show increased upwelling (MERRA-2, JRA-3Q) or zero trends (ERA5). This is in very good agreement with the results from Abalos et al. (2015), who compared three different estimates of upwelling for each reanalysis. Overall, the magnitude of the upwelling trends in Fig. 10 show large differences between the reanalyses, with MERRA-2 being the most negative and ERA5 being the most positive. However, the change of the trend
between the time periods is very similar for all data sets and consistent with the shift in temperature trends.

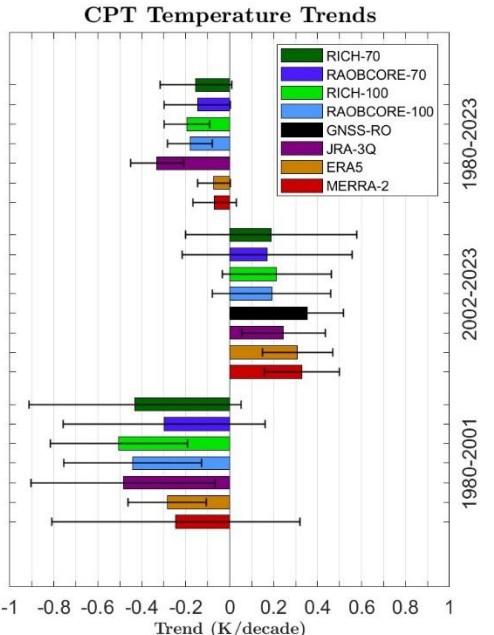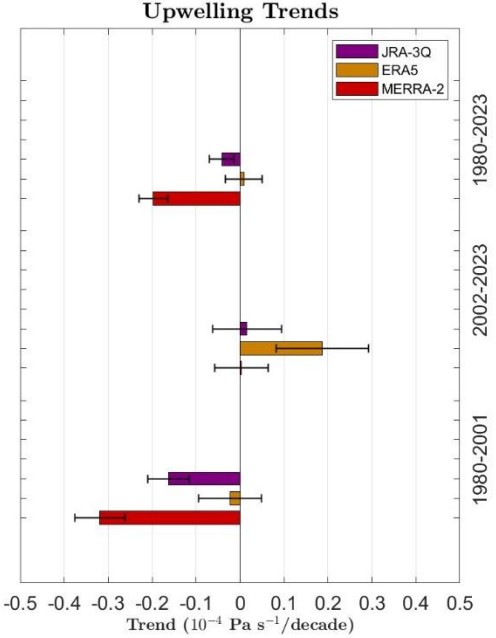

**Figure 10**. Tropical average (30°S-30°N) of trends in cold-point tropopause (CPT) temperature and tropical upwelling based on different observational and reanalysis datasets for the three time periods: 1980-2001, 2002-2023, and 1980-2023. The reanalysis tropical upwelling is the vertical residual circulation averaged over 100 and 70 hPa. Error bars show the ± 2 sigma confidence intervals of the trend taking into account adjusted effective sample size.

In a last step, we compared the seasonal cycle of the cold-point trends with the seasonal cycle of upwelling trends (Fig. 11). If the changes in the BDC are driving the temperature trends, one can expect trends in both variables to have the same seasonality. Cold-point temperature trends from GNSS-RO show a clear seasonal cycle with stronger warming during NH winter and in October. The reanalyses mostly reproduce this seasonal cycle with ERA5 overestimating the amplitude, while MERRA-2 shows a weaker seasonal signal.

The comparison of the seasonal cycle in temperature and upwelling trends show good agreement for ERA5. Strong decrease in upwelling during NH winter as well as in September and October occur at the same time of year as strong cold-point warming. The months of November and December are a bit of an exception to this pattern as upwelling increases or is close to zero and the temperature continues to show positive trends. Furthermore, a spike in the decrease in upwelling in June somewhat obscures the seasonal signal in the upwelling trends. Overall, the similarity of the seasonal cycles of ERA5 upwelling and temperatures trends supports the idea that the temperature trends are at least partially in response to dynamical changes. Upwelling and temperature trends in MERRA-2 do not share a common seasonality with the temperature trend having a relatively flat seasonal cycle while the seasonal cycle of the upwelling trend is similar to the one found in ERA5. JRA-3Q seasonal cycles are somewhere in between ERA5 and MERRA-2 with some shared variability but also relatively weak upwelling changes.

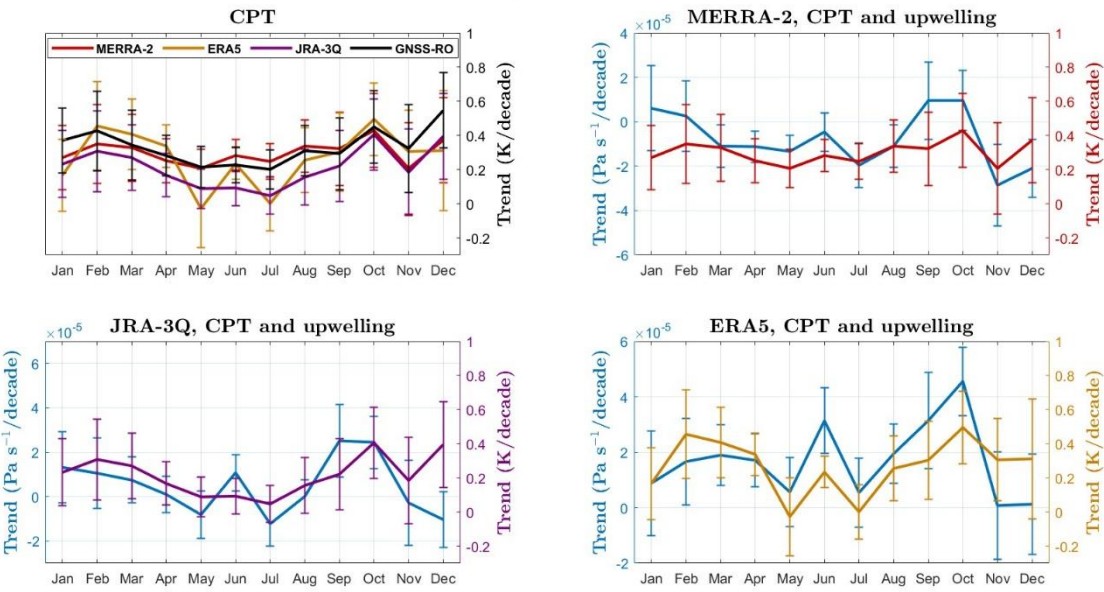

**Figure 11**. Seasonal cycle of tropical trends average (30°S-30°N) in cold-point tropopause (CPT) temperature and tropical upwelling based on different observational and reanalysis datasets for 2002-2023. The reanalysis tropical upwelling is the vertical residual circulation averaged over 100 and 70 hPa. Error bars show the $\pm 2$ sigma confidence intervals of the trend taking into account adjusted effective sample size.

## Conclusion

A significant shift in tropical cold-point temperature trends from cooling in 1980-2001 to warming in 2002-2023 is found in observations. A comparison of the cold-point temperature changes with temperature trends in the upper troposphere and lower stratosphere suggests that the pre-2000 cold-point cooling was driven by ODS-induced upwelling changes similar to what happened in the lower stratosphere. At the same time, it is possible that this cold-point cooling was slightly weakened by GHG-induced warming. Post-2000 cold-point warming, on the other hand, is most likely driven by radiative warming due to

increasing GHGs with some contributions from dynamical changes. The shift in cold-point temperature trends around 2000 points to a regime shift in mechanisms from ODS-induced upwelling changes to GHG-induced radiative warming with some dynamical contributions. While the role of dynamical changes after 2000 is not quite clear, this regime shift illustrates that in the absence of strong upwelling changes, radiative warming could dominate the cold-point temperature trends and thus stratospheric water vapor entry values.

Reanalysis data sets reproduce the robust change in the tropical tropopause temperature trends. The trends in upwelling from reanalyses differ significantly from each other. However, for two reanalysis the upwelling trends are consistent with the changes of cold-point temperatures and show a shift from increased upwelling before 2001 to zero upwelling after 2001. This implies that while the absolute vales of reanalyses upwelling trends remain poorly constrained, their decadal and multidecadal variability is dynamically consistent with changes in temperature.

The here identified changes highlight a strong decadal to multidecadal variability of TTL temperatures which could be related to internal variability of the climate system. Since cold-point temperature trends largely control the amount of water vapor entering the stratosphere, they indirectly influence stratospheric chemistry, radiation and surface climate. CMIP6 models are known to have difficulties in the TTL region and a poor representations of stratospheric water vapor (e.g., Keeble et al., 2021). Quantifying and understanding the decadal variability and long-term TTL temperature trends found in observations can help

to evaluate and improve the representation of the stratosphere in models.

*Data Availability*: Unadjusted IGRA radiosonde data are accessible via the NOAA Integrated Global Radiosonde Archive website at https://www.ncei.noaa.gov/products/weather-balloon/integrated-global-radiosonde-archive (last access: January 2025; Durre et al., 2006). Adjusted radiosonde data can be obtained from the University of Vienna RAOBCORE website at

https://imgw.univie.ac.at/forschung/klimadiagnose/raobcore/ (last access: January 2025; Haimberger, 2007; Haimberger, 2008). GNSS-RO data are available through the COSMIC Data Processing Center at https://www.cosmic.ucar.edu/what-we-do/data-processing-center/data (last access: January 2025). For the reanalysis data, we utilized monthly averaged MERRA-2 temperature fields on pressure levels (Global Modeling and Assimilation Office, 2015). Additionally, we incorporated monthly mean ERA5 temperature data (Copernicus Climate Change Service, Climate Data Store, 2023) and JRA-3Q monthly mean

data (Numerical Prediction Division, Information Infrastructure Department, 2022).

*Author contribution*: MZ performed the analysis and wrote the manuscript. ST developed the research question and guided the research process. RPK provided the GNSS-RO data. SMD provided the reanalysis data. LH provided ROABCORE and RICH data. All authors took part in the process of manuscript preparation.


*Competing interests*: The contact author has declared that none of the authors has any competing interests.

*Financial support*: This research has been supported by the Natural Sciences and Engineering Research Council of Canada (NSERC, grant no. RGPIN-2020-06292).

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

**Appendix A**

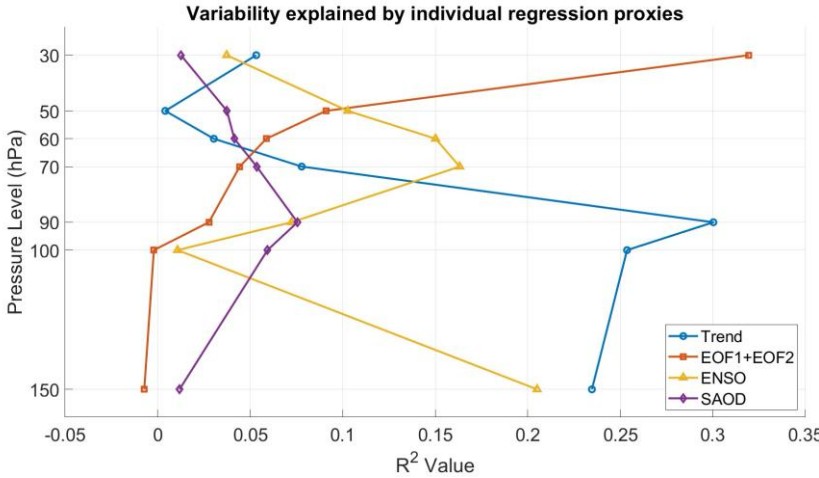

**Figure A1**. Variance in tropical temperature explained by each regressor term (R-square value) for the tropical mean GNSS-RO data 2022-2023.

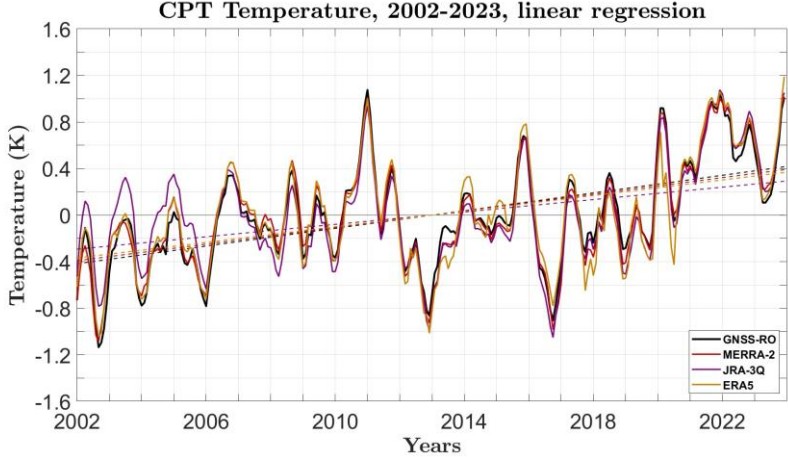

**Figure A2**. Same as Figure 3 but based on a simple conventional linear regression with constant and linear term only

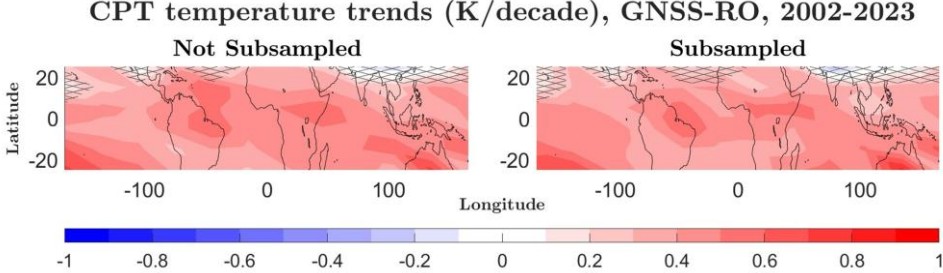

**Figure A3**. Long-term (2002–2023) temperature trends at the cold-point tropopause (CPT) based on (a) regular GNSS-RO and (b) subsampled GNSS-RO data. Trends not significant at the 95% confidence level are marked with grey cross-hatching

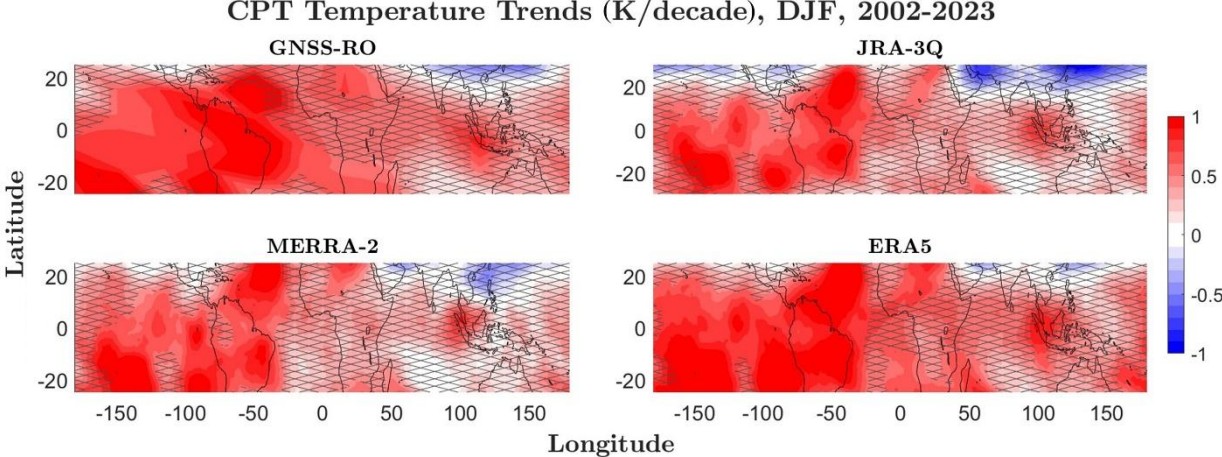

**Figure A4**. Same as the Figure 5 but for DJF.

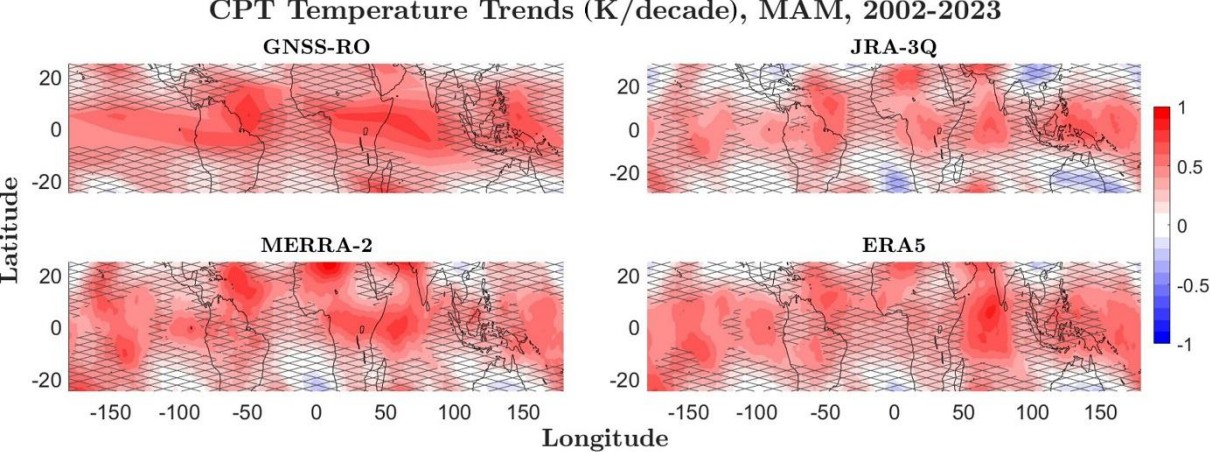

Figure A5. Same as the Figure 5 but for MAM.

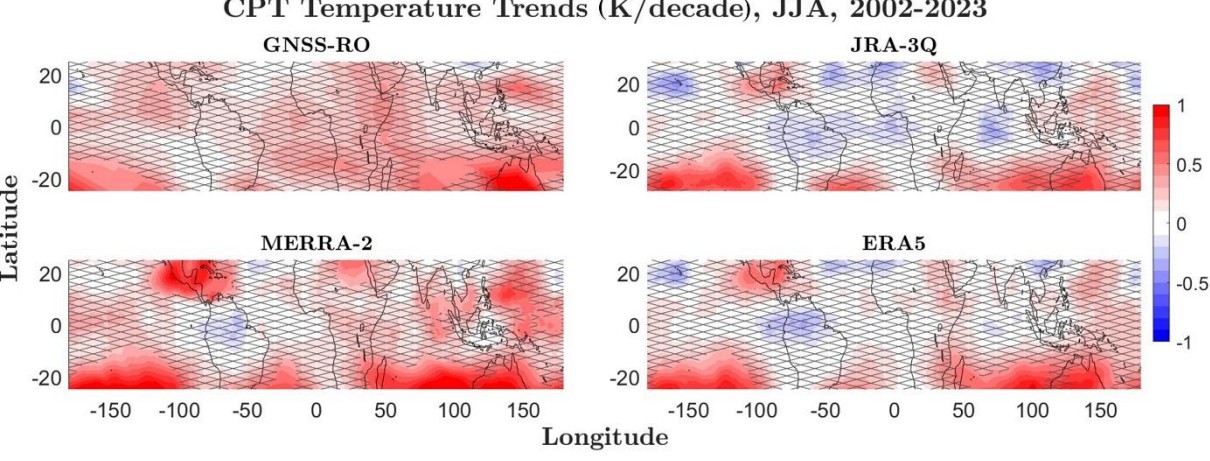

**Figure A6.** Same as the Figure 5 but for JJA.

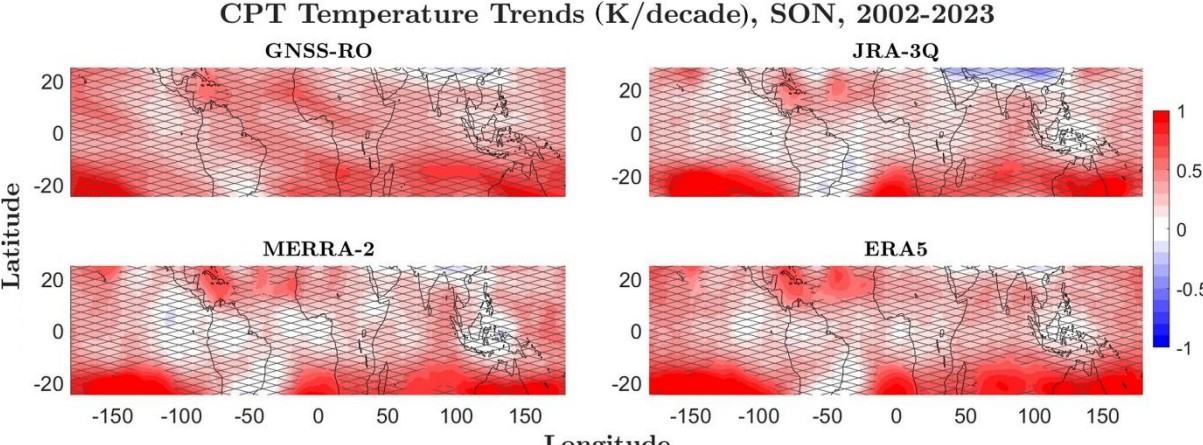

**Figure A7**. Same as the Figure 5 but for SON.

750