# Peer review of "Shift in cold-point tropopause trends derived from radiosonde, satellite, and reanalysis data"

_EGUsphere, 2025_

## Author Comment (AC1)

Author response to referee comments on

"Shift in cold-point tropopause trends derived from radiosonde, satellite, and reanalysis data

by Zolghadrshojaee et al.

**Anonymous Referee #1**

The authors employed radiosonde, GNSS-RO, and reanalyses to update the long-term variations of temperature in the tropical upper troposphere and lower stratosphere, with a particular focus on cold point temperature. As the authors mention in the introduction, there are already many previous studies that use various data sources, including reanalysis, radiosonde, and GNSS-RO. However, a detailed comparison with previous studies is necessary to highlight the novelty of the present study. Furthermore, while the authors attempted to conduct their research using multiple datasets with high vertical resolution, they did not reveal novel or more refined features, nor did they propose new mechanisms for existing trends. Despite the study's objective to quantify trends in both space and time, it does not address the scientific questions that exist in this area. The current state of the manuscript lacks sufficient novelty for publication in the Atmospheric Chemistry and Physics.

We thank the referee for their valuable comments. We have changed the manuscript according to the comments listed below. Most importantly, we have added a detailed discussion of the different potential mechanisms and implications of the shift in temperature trends in the TTL and the role of upwelling. We have also added information on the explained temperature variability and a sensitivity study analyzing sampling density and choice of regression model. As a result of these and other changes, the manuscript has clearly improved and is addressing the scientific questions in the field. Comments are reproduced below, followed by our responses in blue font.

**Major Comments**

1.  As suggested by Highwood and Hoskins (1998), the cold-point is only a reliable tropopause definition when the lower stratosphere is not close to being isothermal, i.e., within the deep tropics. Beyond the deep tropics, the validity of the physical significance of cold point identified in the research region [30S-30N] must be assured.

    Thanks for pointing this out. We prefer to keep the region as 30S-30N in order to include the summer monsoon regions of the Northern Hemisphere. Air masses can enter the stratosphere via the Asian or North American summer monsoon circulation (e.g., Figure 5 in Fueglistaler et al., 2005; Yan et al., 2019) and therefore the coldest temperatures in these regions and their trends are relevant for stratospheric water vapor abundances.
    We have analysed temperature profiles averaged over the inner tropics as well as over 20N-30N and 20S-30S to check that the latter two regions are not close to being isothermal and that the cold point can be determined. As the Figure below illustrates the temperature gradients in the 20N-30N and 20S-30S region are not as steep as in the inner tropics but nevertheless pronounced enough to determine the cold point tropopause.

[Figure]

2. What are the differences between the trend term derived from multivariate linear regression and that derived from conventional linear regression? Furthermore, it is crucial to ascertain the magnitude of the contributions of the additional terms (QBO, ENSO, SAOD) in the multivariate linear regression.

Differences between trends derived from multivariate linear regression and from a conventional linear regression are less than 10%. The table below lists the cold point trends for GNSS-RO, MERRA-2, JRA-55 and ERA5 based on a multivariate linear regression including QBO, ENSO, SAOD and based on a conventional linear regression with trend terms being very similar for both methods. We have added this information to the manuscript (2nd paragraph of section 4) and a figure with trends based on the conventional linear regression to the Appendix A (Fig. A2).

[Figure]

We have also included information on the variance in tropical temperature explained by each regressor term for the tropical mean GNSS-RO data 2022-2023. The figure below gives the R-square value (coefficient of determination) of the regression analysis if the regression only includes a constant and one independent variable (trend, QBO, ENSO and SAOD, respectively). The proportions of variance in the dependent variable (temperature) that can be explained by the individual independent variables are relatively low. In the upper troposphere and TTL, the trend term captures most of the variability as

the temperature shows strong positive trends at these levels. ENSO is important in the upper troposphere while the QBO explains temperature variability more in the lower to middle stratosphere (above 50 hPa). The impact of the SAOD is relatively low throughout the region shown here. We have added the figure to the Appendix A (Fig. A1) and a short comment to the first paragraph of Section 4.

[Figure]

3. GNSS-RO measurements exhibit pronounced irregularities in the tropics compared to middle latitudes. The authors use a relatively large grid (30° longitude × 10° latitude) to ensure sufficient data sampling. However, early single GNSS-RO missions were capable of providing only about 100 profiles per day globally. For instance, the CHAMP (2001-2007) and GRACE (2006-2007) satellites each provided approximately 130 daily profiles. By contrast, the COSMIC constellation, operational since late 2007, provides about 2,500 measurements daily. This substantial disparity highlights significant sampling inhomogeneity. Given the use of multiple missions to construct long-term observations, how do the authors account for sampling biases arising from mission prioritization?

This is a good question, and we have investigated the impact of lower sampling density in a small case study now added to the manuscript. In addition, it is important to note that as a self-calibrating technique, GNSS-RO data show excellent consistency, mission independence and good precision among the individual satellite missions. In consequence, GNSS-RO data have been used to characterize long-term radiosonde biases in the UTLS (Ho et al., 2017) and are ideal candidates for the use as climate data record (Steiner et al., 2020).

In order to test the impact of the varying sampling densities across the different missions on spatially resolved temperature trends, we have resampled the GNSS-RO data for 2006-2024. The resampling was carried out by capping the number of profiles from all missions to 6000 a month globally to match the sampling density of the CHAMP 2002-2005 mission. For each month, the 6000 profiles were chosen randomly from all available profiles. The spatially resolved cold-point trends based on the subsampled data (right panel of figure below) are very similar to the original trends based on unsampled data (left panel in figure below). While some smaller differences can be seen, the magnitude and distribution of the trends show overall excellent agreement suggesting that the sampling only has a small impact on the results presented in our manuscript. We have added this information to the manuscript (5[th]

paragraph of section 4) and the figure with trends based on the conventional linear regression to the Appendix A (Fig. A3).

We have also updated all figures containing GNSS-RO data after detecting an error in the processing code. Results have not changed, and the figures are nearly identical when compared to their old versions.

[Figure]

4. What implications might be drawn from the observation of significant upper troposphere warming in conjunction with insignificant temperature trends in both the cold point and lower stratosphere, as illustrated in Figure 6? Furthermore, it is important to consider the implications of these phenomena in relation to radiative or radiative-dynamical balances.

We have extended the conclusion section with a discussion of the implications of the shift in cold-point temperature trends. Based on the comparison of the cold-point temperature changes with temperature trends in the upper troposphere and lower stratosphere, we can hypothesize that the pre-2000 cold-point cooling was driven by both GHG- and ODS-induced upwelling changes. At the same time, it is possible that this cold-point cooling was slightly weakened by GHG-induced warming. Post-2000 cold-point warming on the other hand, is most likely driven by radiative warming due to increasing GHGs with some contribution from upwelling changes. The shift in cold-point temperature trends around 2000 points to a regime shift in mechanism, from ODS-induced dynamical changes to GHG-induced radiative warming with some dynamical contributions. While the role of dynamical changes after 2000 is not quite clear, this regime shift illustrates that in the absence of strong dynamical upwelling changes, radiative warming will dominate the cold-point temperature trends and thus stratospheric water vapor entry values.

5. A discussion is also required regarding the discrepancy between the trends derived from different datasets.

We have added a discussion of these discrepancies to the manuscript (3rd and 4[th] paragraph of section 4) including aspects of bias corrections of observations assimilated in reanalysis and homogenization approaches.

6. An important question concerns the apparent shift in long-term trends around the year 2000. While the authors have proposed several potential mechanisms for this shift, a more thorough discussion is needed to fully explore these possibilities.

We have added a detailed discussion (3 new paragraphs in Section 6) of the different mechanisms that can impact the shift in temperature trends in the TTL. Based on model derived profiles of temperature trends due to increasing GHGs and changing ODSs available in the literature and other publications,

we now discuss the potential contributions of changes in ODSs, ozone, GHGs and dynamics to TTL temperature trends before and after 2000.

7. Given the apparent decoupling of upwelling trends and warming trends, it is necessary to explore alternative mechanisms that could explain the observed changes. The authors propose a correlation between the warming of the cold point and the weakening of BDC (Ln 305). This relationship should be quantitatively assessed through a rigorous analysis of reanalysis data.

Upwelling trends in the three reanalysis are largely consistent with the temperature trend shift around 2000. More upwelling in MERRA-2 and JRA-3Q before 2000 is consistent with cold-point cooling over the same time period. This change in upwelling is also consistent with a model attribution study suggesting that lower stratospheric cooling before 2000 was driven by an ODS-induced increase in upwelling (Polvani et al., 2017). Upwelling trends close to zero in MERRA-2 and JRA-3Q after 2000 suggest that cold-point warming after 2000 is not driven by dynamical changes but by radiative effects of increasing GHGs. Upwelling trends in ERA-5, on the other hand, suggest decreased upwelling after 2000 as a potential driver of the warming, while no upwelling changes before 2000 imply that the cooling trends over this time were driven by direct radiative effects. We have substantially extended the discussion of these results in Section 6 to make this clearer.

8. Regarding the zonal-mean dynamical variables from Martineau et al. (2018) (covering 1958-2016), clarification is needed as to whether the vertical residual velocities were: (1) calculated following Martineau's methodology, or (2) obtained directly from the archived data. The relevant description in Lines 184-186 should be revised to clarify this point, or alternatively, the Data Availability statement should be updated accordingly.

Thanks for pointing this out. We have obtained the variables directly from the archived data. We have corrected the sentence, which now states: 'Reanalysis estimates of the residual velocity were taken from the Atmospheric Processes And their Role in Climate (APARC) Reanalysis Intercomparison Project (A-RIP) zonal mean data set Martineau et al. (2018).'

**Minor comments**

Section 2.1 requires rewording. The methodology suggests that the cold point is determined for each radiosonde profile prior to the gridding procedure.

The Section has been rephrased.

Ln 152-154 Please rephrase this sentence
The sentence has been rephrased.

Ln 158 balloon reading?
The sentence has been rephrased.

Ln 175 It is necessary to establish the reasons for the observation of a comparatively wider range of cold points in reanalyses as compared with radiosonde observations.
The boundaries were chosen so that they could be applied globally (i.e., at all latitudes) and for all seasons. The 10 hPa boundary was originally added to avoid the (very rare) case of having a cold point found in the mesosphere, which can occur during some seasons at higher latitudes. Similarly, the 500 hPa boundary was chosen to make sure and allow for a relatively low altitude cold point for high latitude profiles. For the tropical latitudes considered here, the cold point is always ~80 hPa, and the values we identify are insensitive to the exact choice of pressure boundaries. We have added the information that this calculation was done globally to the manuscript.

Ln 194 In fact, the short-term variations are left in the error term
Correct. However, here we mean the short-term variations explained by known atmospheric modes of variability such as the QBO. We have adjusted the sentence.

It appears that reference to Table 1 has not been made in Ln 326-332.
We have added a reference to Figure 7 to the paragraph.

**References**

Fueglistaler, S., M. Bonazzola, P. H. Haynes, and T. Peter (2005), Stratospheric water vapor predicted from the Lagrangian temperature history of air entering the stratosphere in the tropics, J. Geophys. Res., 110, D08107, doi:10.1029/2004JD005516.

Ho, S.-P., Peng, L., and Vömel, H.: Characterization of the long-term radiosonde temperature biases in the upper troposphere and lower stratosphere using COSMIC and Metop-A/GRAS data from 2006 to 2014, Atmos. Chem. Phys., 17, 4493–4511, https://doi.org/10.5194/acp-17-4493-2017, 2017.

Polvani, L. M., Wang, L., Aquila, V., and Waugh, D. W.: The Impact of Ozone-Depleting Substances on Tropical Upwelling, as Revealed by the Absence of Lower-Stratospheric Cooling since the Late 1990s, J. Climate, 30, 2523–2534, https://doi.org/10.1175/JCLI-D-16-0532.1, 2017.

Steiner, A. K., Ladstädter, F., Ao, C. O., Gleisner, H., Ho, S.-P., Hunt, D., Schmidt, T., Foelsche, U., Kirchengast, G., Kuo, Y.-H., Lauritsen, K. B., Mannucci, A. J., Nielsen, J. K., Schreiner, W., Schwärz, M., Sokolovskiy, S., Syndergaard, S., and Wickert, J.: Consistency and structural uncertainty of multi-mission GPS radio occultation records, Atmos. Meas. Tech., 13, 2547–2575, https://doi.org/10.5194/amt-13-2547-2020, 2020.

Yan, X., Konopka, P., Ploeger, F., Podglajen, A., Wright, J. S., Müller, R., and Riese, M.: The efficiency of transport into the stratosphere via the Asian and North American summer monsoon circulations, Atmos. Chem. Phys., 19, 15629–15649, https://doi.org/10.5194/acp-19-15629-2019, 2019.

---

## Author Comment (AC2)

**Author response to referee comments on**

**"Shift in cold-point tropopause trends derived from radiosonde, satellite, and reanalysis data**

**by Zolghadrshojaee et al.**

**Anonymous Referee #2**

In this study temperature trends are derived from radiosonde, satellite and reanalysis data with special focus on the cold point tropopause. Thereby, the period for which the trend is estimated is extended compared to previous study and covers the time period from 1980-2023, thus more than 40 years. Although this study does not present any significant new results it merits in my opinion publication. The extension of the data considered and the according update in trend estimates justifies a publication since if I understand the summary of previous studies correctly the time periods previously considered where much shorter. Thus, analyzing here about 40 years of data is quite valuable for deriving reliable trend estimates. However, the writing of the manuscript should be improved to better point out what the highlights and major results of this study are.

We thank the referee for their valuable comments. We have changed the manuscript according to the comments listed below. Most importantly, we have rewritten the abstract, shortened some of the data sections and streamlined the discussion of the different potential mechanisms and implications of the shift in temperature trends in the TTL. We have also added information on the explained temperature variability and a sensitivity study analyzing sampling density and choice of regression model. As a result of these and other changes, the manuscript has clearly improved now better highlighting the major results of the study. Comments are reproduced below, followed by our responses in blue font.

**Specific comments:**
Abstract: The abstract needs to be improved. For example, the second paragraph is providing too many details on the results without making clear what the main result is and what the implications of this study are.

> Thanks for pointing this out. We have completely rewritten the second paragraph of the abstract, which is now more focused on the main results as well as potential mechanisms and implications.

P3, L79: You consider here a time period of 40 years! This could be more clearly be pointed out. Especially since it seems that earlier studies considered significantly shorter time periods which were considering max ~ 20 years.

> We have added this fact here and in other places.

P3, L84: The authors published last year a paper on this topic. It would be worth to also clearly point out what the differences between your previous study and this study are.

> We have added one sentence here highlighting the differences between the two studies.

P3, L86: Section 2, especially the sections about the radiosonde data could be improved. I had trouble to understand what the differences between the adjusted and unadjusted data is. I had the feeling that you here also got lost a bit in the details. I would suggest to put 2.1 and 2.2 in one subsection (thus 2.1 and called "Radiosonde data") and then start with an introductory sentence and then have two subsubsection headers (with or without section numbering) on the adjusted and unadjusted radiosonde data.

> Thanks for the suggestion, we have restructured sections 2.1 and 2.2 accordingly and have added short summary of the differences between the two data types.

P4, L116ff: Does this paragraph really belong to this subsection? I had rather the feeling this is part of the result section.

> Given that the very good agreement of the residuals from adjusted and unadjusted data at the fixed pressure levels is an important prerequisite for our methodology, we prefer to show this figure before the method section.

P4, L120: Has the abbreviation "QBO" been introduced?

> The name has been added.

P6, 158-170: Is this information really necessary. I have the feeling also here too many details are provided which are not important for understanding this study.

> We have considerably shortened this paragraph.

P6, L171-181: I think it would be much more helpful to list these differences in a table.

> We have considerably shortened and restructured this section, but decided to not introduce a table given that there are only 3 quantities that would be included in such as table.

P5-7: The reanalysis data section is too long and I had the feeling too many results are presented here that are not necessary for understanding your study.

> We have considerably shortened this section by removing some details.

P8, L219-220: If in both cases the same pressure levels have been used the sentence could be formulated much shorter and clearer.

> I can't really see how this sentence could be formulated much shorter.

P8, L234-236: The latter part in the sentence "The good agreement of the four trend estimates highlights the consistency of the CPT temperature trends derived from adjusted data sets, which are only half of the cooling trend suggested by unadjusted data." not clear. Please rephrase.

> We have rephrased the sentence as 'The good agreement of the four trend estimates based on adjusted data highlights the consistency of our method when applied to different pressure levels or data sets. These adjusted trends are only half of the cooling trend suggested by unadjusted data, reaffirming that inhomogeneities in the latter can impact the trend estimates.'

P9, L244: Add "e.g." before the reference "Randel et al.".

> Done.

P16, L374: Add here one or two sentences summarizing what has been done in this study.

> We have added one sentence summarizing what has been done.

P17 , L387-389: This is an important result and should be mentioned at several places in the manuscript and not solely in the middle of the discussion.

> We have now highlighted this shift in cold-point trends in other parts of the manuscript such as the abstract and overall conclusion section. We also extended the conclusion section with a discussion of the different mechanisms that can impact the shift in temperature trends in the TTL and possible implications

All figures: Put the units in the title in parentheses instead of brackets.

> Done.

**Technical corrections:**
P6, L173: space between number and unit missing.
P8, L224: space between number and unit missing.
P11, L283: in the Appendix A section -> in Appendix A
P19, L445: ozon e -> ozone

P25, Appendix: Start with the Appendix on a new page.

All changes have been made.

---

## Author Response (AR2)

*Dear Editor,*

*We express our gratitude to the editor for their invaluable comments, Comments are reproduced below, followed by our responses in italics.*

Specific comments:
P1, L10ff: The abstract is too long and needs to be shortened (see the new ACP author guidelines: https://www.atmospheric-chemistry-and-physics.net/policies/guidelines_for_authors.html, maximum allowed length is 250 words).

> *Here is the shortend abstract: The tropical tropopause layer is the transition region between the well-mixed convective troposphere and the radiatively controlled stratosphere and plays a crucial role for air mass transport between these layers. In this paper, we present updated trends of tropopause and lower stratospheric temperature from radiosondes, GNSS-RO data and the reanalyses ERA5, JRA-3Q, and MERRA-2. Given its importance in determining the concentration of water vapor entering the stratosphere, we focus on temperature trends at the cold-point tropopause, which we determined from radiosonde observations after correcting for time-varying biases.*

> *Radiosonde and GNSS-RO data show a significant shift from strong cold-point cooling for 1980-2001 to warming for 2002-2023. Reanalysis data sets reproduce the robust change in the tropical tropopause temperature trends and furthermore show opposite trends in tropical upwelling for 1980-2001 compared to 2002-2023, consistent with the cold-point and lower stratosphere temperature changes. The shift in cold-point temperature trends around 2000 suggests a regime shift in the dominant mechanism controlling CPT temperatures, from ozone depleting substance-induced dynamical changes before 2000 to greenhouse gas-induced radiative warming with some dynamical contributions after 2000. While the role of dynamical changes after 2000 is not completely clear, this regime shift suggests that in the absence of strong dynamically induced cooling trends, radiative warming could dominate the cold-point temperature trends and thus stratospheric water vapor entry values.*

P2, L59-60: Please rephrase the sentence so that it becomes clear that Aura is the satellite and MLS the instrument onboard of it. A suggestion would be "…as detected by the Microwave Limb Sounder (MLS) onboard the NASA's Aura satellite."

P3, L81: I would suggest to rephrase the sentence as follows: "In our study, we aimed on updating existing studies of…….".

P3, L82: I would suggest to write instead of "Our analysis will focus…." rather "Our analysis focuses on……"

P3, L87ff: "Section" should be abbreviated as "Sect." unless it appears at the begin of the sentence (please correct this throughout the manuscript).

P4, L98: Add a reference for this statement?

P4, L107: Time is usually abbreviated with a small "t", thus I would suggest to write "t00" or "t0" and "t12".

P4, L115: The dataset? You mean the IGRA dataset. I would suggest to clearly state this.

P5, Figure 1 caption and all following figure captions: Only the text "Figure" with the respective number is written with bold face. The rest of the caption should have the same font as the text.

P6, L150-151: Abbreviations of the instrument names should be introduced and "Metop" should be written consistently as "Metop" or "METOP" (I think the correct writing is "Metop").

P6, L162: Abbreviation "ECMWF" should be introduced.

P6, L167: This was rather an "update" of the current data set than a "new" data set.

P6, L173-175: Also here the instruments abbreviations should be introduced.

P7, L198, Sect. 3.1 title: error bars -> errors

P7, L210: Add here the also the abbreviation of this data set in parenthesis -> (GloSSAC)

P8, L215: Write here in the section header Nearby Level also with quotation marks -> 'Nearby Level'

P10, L274: I would suggest to move "with differences of less than 10%" behind "very similar" and using parenthesis.

P14, Figure 6 caption: Rather "hatches" than "crosses"?

P17, L394: Is this shown in this study? Which section? Add a reference or on which place in the manuscript this is discussed.

*For all the above comments, we have modified the text accordingly.*

In Line 389 you mention a warming, but without a specific month

*We have modified the next paragraph and mentioned October warming as well:*

*"The seasonal cycle of the temperature trends shows that the 1980-2001 lower stratospheric cooling is only interrupted in March when temperature trends are around zero for the radiosonde data. The absence of cooling in March might be related to the weakening of the BDC in this month, as the Arctic polar vortex has intensified during NH spring over 1979-1999 (Langematz and Kunze, 2006). This signature of the BDC trends in the seasonal cycle of the tropical cold-point temperature trends with a reduced cooling*

*trend in March (end of NH winter) is similar to the strong October warming found for 2002-2023 at the end of the SH winter (Fig. 4).”*

P18, Figure 9 caption: Same here as for the caption of Figure 6. Rather "hatches" than "crosses".

P17, L405: "Discussion and reanalysis vertical residual circulation" sounds weird. I would suggest to name this section just "Discussion".
P18, L429: It is not clear if you refer here to Fig. 1 in your manuscript or in Shine et al. (2003).

P19, L440: Add a reference for this statement
P19, L444: "Here" not clear. Do you mean in this study or in this section?References: Journal names should be abbreviated. For some references it is done, but for some not. Note, for the reference of Simmons (2020) the journal and volume are missing.
Technical corrections:
P3, L82: Add "by" so that it reads "by covering".
P3, L93: "last" -> "past" (?)
P7, L180: skip "models"
P7, L185: ERA-5 -> ERA5
P10, L274: omit "Appendix", just writing "see Fig. A2" is sufficient.
P12, L307: omit also here "Appendix".
P12, Figure 5 caption: MERRA-22 -> MERRA-2
P18, L424: It should read "Arblaster et al, 2014" and a space between the year and "and" is missing.
P18, L435: Add "by" so that it reads "……by somewhat weakening…".

P19, L462: shows -> show

P20, L477: shows -> show

P20, Figure 10 caption: for three periods -> for the three time periods

P22, 517-521: Rather than "last accessed" I would suggest to write "last access".

*For all the above comments, we have modified the text accordingly.*